# CauFR-TS: Causal Time-Series Identifiability via Factorized Representations

**Ayanabha Ghosh**[*]                                          *p23iot002@iitj.ac.in*
*Centre for AIoT and Applications*
*Indian Institute of Technology Jodhpur*

**Debasis Das**                                                *debasis@iitj.ac.in*
*Department of Computer Science and Engineering*
*Indian Institute of Technology Jodhpur*

**Asif Ekbal**                                                 *asif@iitp.ac.in*
*Department of Computer Science and Engineering*
*Indian Institute of Technology Patna*

**Reviewed on OpenReview:** *https://openreview.net/forum?id=Al4OnLoQsp*

## Abstract

Causal discovery from multivariate time series is a fundamental problem for interpretable modelling, causality-aware downstream analysis, and intervention-driven simulation. Recent neural approaches commonly rely on shared latent embeddings to capture temporal dynamics and utilize them for causal structure estimation and downstream prediction. We formally establish that such shared encoders entangle distinct causal mechanisms into a unified latent manifold, which exhibits fundamental theoretical limitations of structural non-identifiability and conditional independence assumptions required for Granger causality. To address these issues, we propose CauFR-TS, a recurrent variational framework that enforces mechanism modularity through dimension-wise encoders and ensures mediation of all cross-variable dependencies through structured latent aggregation. Furthermore, we address the instability of heuristic thresholding in continuous relaxation methods by proposing an adaptive, data-driven unsupervised link selection strategy based on decoder weight distribution. Empirical evaluation on synthetic and in silico biological benchmarks demonstrates that CauFR-TS outperforms recent baselines in graph recovery metrics while preserving competitive probabilistic forecasting performance. The implementation is available at: https://github.com/ayan-cs/caufr-ts.

## 1 Introduction

Understanding causal relationships in multivariate time series is a fundamental problem with implications for forecasting, decision-making, and intervention-aware simulation in domains such as climate science (Fizaine & Le Borgne, 2025), healthcare (Feuerriegel et al., 2024), and industrial monitoring (Ma et al., 2025). Unlike static causal discovery, temporal causal inference must simultaneously account for temporal dependencies, delayed effects, and feedback loops (Mokhtarian et al., 2025; Ferdous et al., 2025) while remaining robust to noise, nonlinearity, and high dimensionality (Aslan & Ombao, 2025; Zhou et al., 2024). As a result, among the various paradigms for time-series causal discovery (Hasan et al., 2023), Granger causality and its nonlinear extensions have emerged as a dominant operational bedrock for causal discovery in time series (Mameche et al., 2025; Gong et al., 2024), positing that a variable $X^j$ causes $X^i$ if the unique history of $X^j$ reduces the predictive error of $X^i$. Constraint-based approaches (Gao et al., 2023; Runge, 2020; Runge

---

[*]Corresponding author

et al., 2019) exploit conditional independence tests to recover causal skeletons from observational data, while score-based methods (Sun et al., 2023; Pamfil et al., 2020) optimize over a space of candidate graph structures using a global criterion. Neural Granger causality methods, in contrast, naturally integrate representation learning with causal structure estimation (Redden et al., 2026), making them particularly suited to nonlinear, high-dimensional temporal settings.

While classical Vector Autoregressive models (Moneta et al., 2011; Haufe et al., 2010) provide interpretability, they fail to capture the nonlinear dynamics inherent in real-world systems. This limitation has spurred a paradigm shift towards neural Granger causality methods, where deep neural networks parameterize autoregressive functions (Lin et al., 2024; Bussmann et al., 2021). Such methods leverage expressive function approximators to model nonlinear temporal dynamics while inducing sparsity in lagged dependencies, enabling scalable causal graph estimation. Subsequent works have extended this paradigm to high-dimensional settings (Mokhtarian et al., 2025), irregularly sampled time series (Cheng et al., 2024), and scenarios incorporating interventional or heterogeneous data (Varıcı et al., 2024). More recent transformer-based approaches (Kong et al., 2025; Huang et al., 2025) further aim to unify temporal representation learning and causal graph estimation within a single end-to-end framework, offering improved performance and interpretability.

Despite their empirical success, we identify a critical theoretical oversight in the design of modern causal time-series models. The prevailing architectural paradigm employs a shared encoder processing the full multivariate input vector $X_t$ to map history into a unified latent representation $Z_t$. This latent code is subsequently reused for forecasting, simulation, and causal structure estimation (Lin et al., 2024; Li et al., 2023). While this design is computationally appealing, we argue that this design fundamentally violates the Principle of Independent Causal Mechanisms (ICM) (Parascandolo et al., 2018). By compressing the history of all variables into a shared vector, the encoder inevitably entangles the information of distinct mechanisms. From a causal inference perspective, this shared representation acts as an unobserved architectural confounder that violates the conditional independence assumptions (Seth & Príncipe, 2012) required for Granger causality. Even if variable $X^j$ has no true causal effect on $X^i$, a shared encoder may leak information about $X^j$ into the latent representation used to predict $X^i$. Consequently, downstream detectors, including attention heads or sparse decoders, operate on confounded representations, rendering the resulting causal graphs structurally non-identifiable (Bodik & Chavez-Demoulin, 2025; Brogueira & Figueiredo, 2025).

Several recent works attempt to mitigate related challenges through architectural regularization, graph priors, or adaptive lag selection (Zoroddu et al., 2024; Zhang et al., 2024). Methods incorporating prior structural knowledge constrain the search space of causal graphs, while flexible time-windowed and dynamic causal discovery approaches address temporal heterogeneity and evolving dependencies. However, these methods typically assume that the learned latent representations are causally valid and do not explicitly prevent cross-variable information sharing at the representation level (Chen et al., 2025; Zhang et al., 2025). Similarly, thresholding strategies for causal link selection are often heuristic or manually tuned, leading to instability across datasets and limiting reproducibility (Lee & Honavar, 2020). Therefore, reliable causal identification in deep time-series models requires explicit enforcement of mechanism modularity at the representation level.

To bridge the gap between deep generative expressivity and rigorous causal identifiability, we propose CauFR-TS. Unlike prior approaches that rely on post-hoc interpretation of entangled representations or solely impose sparsity at the decoder level, CauFR-TS enforces causal modularity directly at the representation level. Specifically, we introduce a factorized encoder architecture in which each time series variable is processed by an independent neural module, thereby strictly preventing information leakage in the latent space. In addition, to address the sensitivity of existing methods to manually tuned thresholding hyperparameters, we propose an adaptive, parameter-free causal thresholding strategy based on a Gaussian Mixture Model of learned decoder weights, which robustly separates causal signals from noise and yields stable causal graphs. We further provide a theoretical analysis showing that CauFR-TS restores causal identifiability under the Principle of Independent Causal Mechanisms. Extensive empirical evaluation on synthetic benchmarks and biologically grounded datasets demonstrate that CauFR-TS consistently outperforms the state-of-the-art baselines in causal graph recovery. Furthermore, we empirically verify that enforcing causal modularity does not degrade predictive performance, thereby decoupling causal identifiability from forecasting accuracy.

## 2   Preliminaries

We briefly introduce the notation, causal notions, and identifiability principles underlying our analysis. These concepts form the theoretical basis for Granger causal discovery in multivariate time series and motivate the architectural considerations examined in the subsequent section.

**Definition 1 (Granger Causality)** *A time series component $x^{(j)}$ is said to Granger-cause $x^{(i)}$ if the prediction of $x_t^{(i)}$ based on the full history $X_{t-}$ is strictly better, in terms of reduced prediction error variance, than the prediction based on the history excluding $x^{(j)}$, denoted as $X_{t-}^{-j}$. Formally, if $P(x_t^{(i)}|X_{t-}) \neq P(x_t^{(i)}|X_{t-}^{-j})$, then a directed Granger-causal link $x^{(j)} \to x^{(i)}$ exists.*

Granger causality (Granger, 1969) operationalizes causality through conditional predictive dependence and is widely adopted in temporal causal discovery. Here, the temporal causal structure is represented by a directed graph $G = (V, E)$ where each node $v \in V$ corresponds to a time series variable and each directed edge $(j, i) \in E$ denotes a Granger-causal influence of $x^{(j)}$ on $x^{(i)}$. This structure is compactly encoded in the causal adjacency matrix $A \in \{0,1\}^{D \times D}$, where $A_{ij} = 1$ iff $x^{(j)}$ Granger causes $x^{(i)}$.

**Definition 2 (Structural Identifiability)** *Let $\mathcal{M} = \{P_\theta : \theta \in \Omega\}$ be a parametric model class over observable data $X$, where $\theta$ parameterizes both the data-generating mechanisms and an associated causal graph $G(\theta)$. The model class $\mathcal{M}$ is said to be structurally identifiable if, for any two parameter sets $\theta, \theta' \in \Omega$, $P_\theta(X) = P_{\theta'}(X) \Rightarrow \theta = \theta'$, up to trivial symmetries such as permutation or reparameterization that do not alter the causal structure.*

In non-linear settings, structural identifiability is generally not guaranteed without additional assumptions. Results from non-linear Independent Component Analysis show that latent representations learned from observational data are typically identifiable only up to complex non-linear transformations, unless constraints, such as sparsity, interventions, or non-stationarity are imposed.

**Definition 3 (Graph Identifiability)** *A causal model class $\mathcal{M}$ is said to be graph identifiable if, for any two parameter sets $\theta, \theta' \in \Omega$, $P_\theta(X) = P_{\theta'}(X) \Rightarrow G(\theta) = G(\theta')$, where $G(\theta)$ denotes the causal graph induced by the parameters $\theta$.*

In causal discovery from observational time series, graph identifiability is a necessary condition for reliable inference. If multiple causal graphs induce the same observational distribution, no algorithm can distinguish them without additional assumptions.

**Definition 4 (Independent Causal Mechanisms (ICM))** *Let $X_t = (x_t^{(1)}, \ldots, x_t^{(D)})$ denote a $D$-dimensional time series generated by a causal process with directed temporal graph $G$. The Principle of Independent Causal Mechanisms posits that each variable $x_t^{(p)}$, $\forall p \in \{1, \ldots, D\}$, is generated by a distinct causal mechanism $x_t^{(p)} = f_p\big(PA(x^{(p)})_{t-}, \varepsilon_t^{(p)}\big)$, where $PA(x^{(p)})_{t-}$ denotes the historical causal parents of $x^{(p)}$, $\varepsilon_t^{(p)}$ is an exogenous noise term, and the mechanisms $\{f_p\}_{p=1}^D$ are statistically independent in the sense that knowledge of one mechanism provides no information about the others.*

Under ICM (Gresele et al., 2021), the conditional transition distribution of the multivariate time series factorizes according to the causal graph as:

$$P(X_t|X_{t-}) = \prod_{p=1}^{D} P(x_t^{(p)}|PA(x^{(p)})_{t-})$$

This factorization expresses mechanism modularity, namely that each variable evolves according to its own autonomous mechanism conditioned solely on its direct causes. In neural time-series models, ICM can be respected when each $P(x_t^{(p)}|\cdot)$ is parameterized independently. However, a shared encoder that maps the multivariate history into a single latent representation $z_t$, induces a common conditioning context across variables. If $z_t$ is not explicitly factorized, information from different channels becomes entangled, rendering the conditional distributions $P(x_t^{(p)}|z_t)$ statistically dependent. This violates ICM and motivates encoder-level modularity as a necessary structural prior for causal identifiability.

# 3 Fundamental Limitations of Shared-Latent Causal TS Models

In recent years, a large class of neural time-series models has emerged that jointly perform generative modelling, forecasting, and causal discovery (Kong et al., 2025; Li et al., 2023; Tank et al., 2021). Most of these approaches adopt a shared-latent architectural paradigm, wherein a single encoder maps the multivariate history into a global latent representation, which is subsequently reused by task-specific decoders to generate predictions and infer causal structure. This design underlines recurrent VAE-based causal models, shared-backbone neural Granger causality methods, and several recent transformer-based formulations.

Despite their empirical success, the theoretical implications of shared latent representations for causal identifiability remain poorly understood. In this section, we analyze this architectural paradigm and show that shared latent encoders fundamentally violate the conditional independence assumptions required for Granger causality. As a result, the induced causal graphs are generally non-identifiable, regardless of the model capacity or decoder-level sparsity constraints.

It is important to clarify that ICM is a principle governing the data-generating process, not a constraint imposed on the latent space itself. The factorized encoder is not an assumption about the data; rather, it is a structural design choice that preserves the ICM factorization through the estimation pipeline. If the latent pathway entangles signals from multiple variables, the downstream causal estimator cannot recover the conditional independence structure of the true data-generating process, even when that process genuinely satisfies ICM and even when decoder-level sparsity is enforced.

## 3.1 Model Class and Problem Setup

Let $X_t = (x_t^{(1)}, \cdots, x_t^{(D)}) \in \mathbb{R}^D$ denote a multivariate time series generated by an unknown causal process. We consider a broad class of neural causal time-series models that employ a shared encoder $z_t = f_\theta(X_{t-})$, where $z_t \in \mathbb{R}^H$ is the latent representation computed from the full multivariate history and reused by one or more decoders for prediction, generation, and causal structure estimation. The encoder parameters are shared across variables, and no explicit constraint enforces variable-wise separation in the latent space. This formulation subsumes recurrent VAE-based architectures, shared-backbone neural Granger causality models, and transformer-based approaches relying on a global latent state.

## 3.2 Entanglement in Shared Latent Encoders

We first show that shared encoders inevitably induce information redundancy and non-identifiability of channel-specific contributions.

**Proposition 1 (Entanglement and Information Mixing)** *Let $\mathbf{z}_t = f_\theta(\mathbf{X}_{t-}) \in \mathbb{R}^H$ be a latent representation produced by a shared encoder with parameters $\theta$, where $f_\theta$ jointly processes the full multivariate history $\mathbf{X}_{t-} = (x_{t-}^{(1)}, \ldots, x_{t-}^{(D)})$. Then, in general, $\mathbf{z}_t$ constitutes a non-separable representation in which information from multiple input channels is entangled across its dimensions. Consequently, there exists no unique decomposition $\mathbf{z}_t = (\mathbf{z}_t^{(1)}, \ldots, \mathbf{z}_t^{(D)})$ such that each component $\mathbf{z}_t^{(p)}$ depends exclusively on the history of variable $x^{(p)}$, $\forall p \in \{1, \cdots, D\}$.*

**Proof sketch.** The shared encoder $f_\theta$ maps the full multivariate history $\mathbf{X}_{t-}$ to a latent representation $\mathbf{z}_t$ using a single parameterization. As a result, each latent dimension $h \in \{1, \cdots, H\}$ is, in general, a function of multiple input channels, implying that information from different variables is mixed across the latent space. Although $f_\theta$ is expressive enough to represent arbitrary nonlinear transformations, this expressiveness leads to non-identifiability in the absence of additional structural constraints. In particular, without architectural modularity or independence-enforcing assumptions, multiple latent representations exist that induce identical predictive distributions while assigning different subsets of latent dimensions to different variables.

Consequently, no unique decomposition of $\mathbf{z}_t$ into variable-specific components is identifiable, establishing that shared latent encoders yield the entangled representations that preclude variable-wise separation.

### 3.3 Violation of Conditional Independence

Having established in Proposition 1 that shared latent encoders produce inherently entangled representations, we next show that such entanglement has direct consequences for causal identification. Specifically, entangled latent variables function as unobserved confounders at the representation level, thereby violating the conditional independence conditions required for valid Granger causal inference.

**Proposition 2 (Latent-Induced Confounding)** *Let* $\mathbf{z}_t = f_\theta(\mathbf{X}_{t-})$ *be a latent representation produced by a shared encoder with a single parameterization over all the input channels. When causal influence on a target variable* $\hat{x}_t^{(q)}$ *is estimated through* $\mathbf{z}_t$, *the representation induces confounding between source variables, thereby violating the conditional independence assumptions required for valid Granger causal inference.*

**Proof sketch.** Granger causality relies on conditional independence: if a variable $x^{(p)}$ does not Granger-cause $x^{(q)}$, then conditioning on the histories of all other variables renders $x^{(p)}$ irrelevant for predicting $x_t^{(q)}$. In shared-latent architectures, however, the encoder $f_\theta$ compresses the entire multivariate history into a single latent representation, causing information from multiple source variables to be jointly encoded.

As a result, $\mathbf{z}_t$ generally contains predictive information about $x_t^{(q)}$ originating from variables other than $x^{(p)}$, even after conditioning. This latent aggregation effectively introduces an unobserved confounder between the source variables at the representation level. Consequently, conditioning on $\mathbf{z}_t$ does not recover the required conditional independence, and estimated causal effects from $x_{t-}^{(p)}$ to $x_t^{(q)}$ are biased by spurious dependencies induced by the shared representation.

We provide the detailed theoretical proofs and discussion of the limitations in Appendix D.

**Remark on the scope of Propositions 1 and 2.** The propositions establish that shared encoders provide no architectural guarantee of variable-wise separation in the latent space. This is distinct from claiming that a shared encoder always fails empirically. A sufficiently constrained shared encoder, for instance, one explicitly regularized toward independence, may partially recover causal structure for specific datasets. However, such recovery is contingent on data geometry and optimization dynamics rather than structural properties of the architecture and cannot be relied upon as a general guarantee. Our proposed CauFR-TS provides this guarantee by architectural construction. The formal basis for these claims has been established in Appendix D via Lemma 1, which shows that the Jacobian of a shared encoder contains no structurally zero entries for generic parameters. The subsequent proofs of Propositions 1 and 2 build upon this result.

## 4 Proposed CauFR-TS Framework

Motivated by the theoretical limitations of shared-latent architectures, we propose CauFR-TS, a neural Granger causal discovery framework based on factorized latent representations. The method trains a recurrent variational forecaster to model the conditional distribution $\hat{p}(\mathbf{x}_T|\mathbf{x}_{1:T-1})$, while enforcing structured sparsity in the decoding stage to recover the underlying causal adjacency matrix.

### 4.1 Model Architecture

Our model comprises a multi-head recurrent encoder and a multi-head fully connected decoder. Given a past observation window of length $\tau$, the model can be expressed as,

$$\hat{x}_T^{(d)} = D_\Theta^{(d)}\left(E_\Phi^{(d)}\left(x_{T-\tau:T-1}\right)\right), \quad \forall d \in \{1, \ldots, D\} \tag{1}$$

where $E_\Phi^{(d)}(\cdot)$ and $D_\Theta^d(\cdot)$ denote the encoder and decoder associated with the $d^{th}$ variable, respectively. We do not restrict the architecture of the encoder to a specific recurrent formulation. Instead, any recurrent encoder that processes the lagged observations $x_{T-\tau:T-1}^{(d)}$ and produces a latent embedding $h^{(d)} \in \mathbb{R}^H$ may be employed. Channel-specific embeddings are subsequently reparameterized (Chung et al., 2015; Fabius et al., 2014) and concatenated to form a joint latent representation $\mathbf{z} \in \mathbb{R}^{D \times H}$, which is then provided as input to the decoder. The multi-head decoder is composed of single-layer fully-connected networks, where each

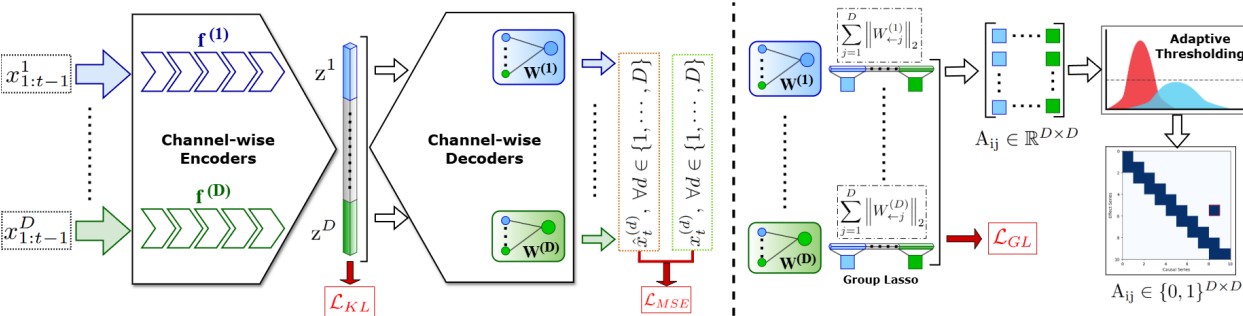

Figure 1: Overview of CauFR-TS. Each variable is encoded independently via a channel-wise encoder, yielding a factorized latent representation **z**. Decoders parameterized by group-sparse weights model cross-channel temporal dependencies. A data-driven thresholding procedure applied to decoder weights recovers the causal adjacency matrix $A$.

decoding head maps the joint latent representation **z** to the parameters of the conditional distribution of $x_T^{(d)}$. Taken together, the encoder-decoder pipeline jointly parameterizes the conditional distribution $p(x_T|x_{1:T-1})$. Subsequently, the prediction for the multivariate observation $x_T$ is constructed by concatenating the outputs of the $D$ decoder heads. A formal description of the encoder and decoder has been provided in Appendix C.

## 4.2 Objective Function

To estimate the causal matrix $A$ during training, we employ the group-wise sparsification method widely adopted in many recent causal discovery works (Rahmani & Frossard, 2025; Dhir et al., 2025; Li et al., 2023). The learning objective of CauFR-TS is to jointly maximize the forecasting accuracy of the multivariate time series while enforcing sparsity in the causal structure, under the assumption that the actual causal matrix $A$ is sparse. We formulate this objective by maximizing the Evidence Lower Bound (ELBO), augmented with a structural sparsity penalty. For optimization, this is implemented as the minimization of a composite loss function comprising the Negative Log-Likelihood and the KL-Divergence. Formally, given a time series instance $x_{1:T-1}$ of $D$ variables, the encoder parameters $\Phi$, and the decoder parameters $\Theta$, the composite objective function can be defined as,

$$\mathcal{L}(\Phi, \Theta) = -\mathbb{E}_{q_\Phi(\mathbf{z}|x)}[\log p_\Theta(x|\mathbf{z})] + \beta \, \mathcal{D}_{KL}(q_\Phi(\mathbf{z}|x)||p(\mathbf{z})) + \lambda \sum_{i=1}^{D} \sum_{j=1}^{D} ||\mathbf{W}_{(j)}^i||_2 \tag{2}$$

where $\beta$ and $\lambda$ are hyperparameters controlling the strength of the variational regularization and the group lasso penalty, respectively. The core mechanism of our causal discovery mechanism is the Group Lasso penalty applied to the decoder weights. Let $\mathbf{W}^i \in \mathbb{R}^{D \cdot K}$ be the weight vector of the decoder head predicting $\hat{x}_t^{(i)}$. We partition $\mathbf{W}^i$ into $D$ groups, where the $j^{th}$ group, denoted as $\mathbf{W}_{(j)}^i \in \mathbb{R}^K$, contains the weights connecting $\mathbf{z}_j$, the latent vector of variable $j$, to the predictor of $i^{th}$ variable. The third term in Equation 2, weighted by $\lambda$, applies an $\ell_1/\ell_2$ regularization in order to introduce the Granger-causal sparsity. If the predictor for the target variable $i$ does not rely on the history of the source variable $j$, the optimization process drives the $\ell_2$ norm of the entire block $\mathbf{W}_{(j)}^i$ to zero. Consequently, the absence of a causal link $j \rightarrow i$ is learned end-to-end, as the decoder effectively blocks the input from $z_j$. In our experiments, we tune $\lambda$ alongside the variational weight $\beta$ to balance identifiability and forecasting performance.

## 4.3 Probabilistic Thresholding of Group-Lasso Weights

We implement a data-driven adaptive thresholding mechanism to eliminate spurious connections and retrieve the causal structure during training. Prior approaches commonly rely on manually specified thresholds on weight norms to identify Granger-causal parents. In this work, we instead model the distribution of learned interaction strengths, measured via the $\ell_2$-norms of decoder weight groups, using a Gaussian Mixture Model (GMM).We fit a two-component GMM to the empirical distribution $p(s)$ of weight norms, modelling

---

**Algorithm 1** CauFR-TS Training Pipeline and Optimization

---

**Require:** Initialized encoder parameters $\Phi = \{\phi_d\}_{d=1}^D$ and decoder parameters $\Theta = \{\theta_d\}_{d=1}^D$; KL and Group
    lasso coefficients $\beta$ & $\lambda$; learning rate $\eta$; Time lag $\tau$;
**Input:** Multivariate time series instance $\{x\}_{t=1}^T$ with $D$ dimensions;
**Output:** Estimated Granger causal matrix $\hat{A}$ and trained model parameters $\{\Phi, \Theta\}$;

1:  **while** not converged or stopping criteria not met **do**
2:     Extract $x_{t-\tau:t-1}$ from $\{x_t\}_{t=1}^T$
3:     **for** $d = 1$ to $D$ **do**
4:         Extract $x_{t-\tau:t-1}^{(d)}$ from $x_{t-\tau:t-1}$
5:         Compute posteriors $(\mu_d, \sigma_d)$ using $\phi_d$
6:         Sample latent $z^{(d)} \sim \mathcal{N}(\mu_d, \mathrm{diag}(\sigma_d^2))$                              $\triangleright \mathcal{L}_{KL}$
7:     **end for**
8:     Construct latent representation $z = \{z^{(d)}\}_{d=1}^D$
9:     **for** $d = 1$ to $D$ **do**
10:       Predict $\hat{x}_t^{(d)}$ from $z$ using $\theta_d$                                    $\triangleright \mathcal{L}_{MSE}$
11:       Extract decoder weights $\mathbf{W}_{(j)}^{(d)}$ connecting $z_j$ to target $d$     $\triangleright \mathcal{L}_{GL}$
12:     **end for**
13:     Compute total loss $\mathcal{L}$ from Equation 2
14:     Update $\{\Phi, \Theta\}$ using AdamW optimizer
15: **end while**
16: Invoke Algorithm 2 to retrieve $\hat{A}$
17: **return** $\hat{A}$ and trained $\{\Phi, \Theta\}$.

---

interaction strengths as a bimodal distribution with one component capturing weak, non-causal interactions and the other capturing strong, potentially causal interactions. Formally, we can define it as,

$$p(s) = \pi_{\mathrm{noise}}\mathcal{N}(s; \mu_{\mathrm{noise}}, \sigma_{\mathrm{noise}}^2) + \pi_{\mathrm{signal}}\mathcal{N}(s; \mu_{\mathrm{signal}}, \sigma_{\mathrm{signal}}^2) \tag{3}$$

where $\pi_k, \mu_k, \sigma_k^2$ represent the mixing coefficient, mean, and variance for component $k \in \{\mathrm{noise}, \mathrm{signal}\}$. We enforce the structural constraint $\mu_{\mathrm{noise}} < \mu_{\mathrm{signal}}$ to identify the components. The parameters $\psi = \{\pi_k, \mu_k, \sigma_k\}_k$ are estimated via the Expectation-Maximization (EM) algorithm. Given the estimated mixture parameters, the optimal threshold $\tau^*$ is defined as the point that minimizes the Bayes classification error, equivalently given by the intersection of the posterior probabilities of the two Gaussian components.

Let $\gamma_k(s) = P(C = k|s)$ be the posterior probability that an observed strength $s$ belongs to component $k$. The decision boundary $\tau^*$ is the value in the interval $[\mu_{\mathrm{noise}}, \mu_{\mathrm{signal}}]$ satisfying:

$$\tau^* = \{s \in \mathbb{R}^+ \mid \gamma_{\mathrm{signal}}(s) = \gamma_{\mathrm{noise}}(s)\} \tag{4}$$

Practically, $\tau^*$ is where the probability density of the causal signal begins to dominate the noise tail. The final binary causal matrix $\hat{A}$ is then obtained as follows,

$$\hat{A}_{ij} = \begin{cases} 1 & \text{if } s_{ij} > \tau^* \\ 0 & \text{otherwise} \end{cases}$$

By adapting $\tau^*$ to the empirical distribution of learned weights, the method reduces reliance on manually specified hyperparameters and exhibits robustness across different scales of causal effects. Empirical validation of the bimodal assumption across all the datasets is provided in Appendix F.3.

## 5 Experimental Setup

In this section, we first outline the datasets, baselines, and evaluation metrics, followed by reporting results on causal discovery, predictive accuracy, and ablation analyses in the next section.

---

**Algorithm 2** Probabilistic Thresholding of Group-Lasso Weights using Gaussian Mixture Model

---

**Require:** Decoder weight groups $\mathbf{W}^i_{(j)}$, $\forall i, j \in \{1, \dots, D\}$

**Ensure:** Binary causal adjacency matrix $\hat{A}$

  1: Initialize set of interaction strengths $\mathcal{S} = \phi$
  2: **for** each variable $i = 1$ to $D$ and each group $j = 1$ to $D$ **do**
  3:     Compute interaction strengths $s^i_{(j)} \leftarrow \|\mathbf{W}^i_{(j)}\|_2$
  4:     $\mathcal{S} \leftarrow \mathcal{S} \cup \{s^i_{(j)}\}$
  5: **end for**
  6: Construct empirical distribution $p(s)$ from $\mathcal{S}$
  7: Fit a two-component GMM to $p(s)$                                     ▷ Equation 3
  8: Compute posteriors $\gamma_k(s) = P(C = k \mid s)$ for $k \in \{\text{noise}, \text{signal}\}$
  9: Identify Noise cluster and Signal cluster where $\mu_{\text{noise}} < \mu_{\text{signal}}$.
10: Determine adaptive threshold $\tau^*$                                 ▷ Equation 4
11: Obtain $\hat{A}_{ij} = \mathbb{I}(\|\mathbf{W}^i_{(j)}\|_2 > \tau^*)$
12: **return** $\hat{A}$

---

## 5.1 Datasets

We systematically evaluate the proposed approach on both causal discovery and time-series forecasting tasks using four synthetic causal time-series benchmarks and two in silico simulated biological datasets, all of which provide access to ground-truth causal graphs.

**Hénon maps.** We consider a system of 10 coupled Hénon chaotic maps (Kugiumtzis, 2013), in which the ground-truth causal structure follows a unidirectional chain $x^{i-1} \rightarrow x^i$. A total of 5,000 samples are generated and used for training and evaluation.

**Lorenz-96 model.** The Lorenz system (Lorenz, 1996) is a nonlinear dynamical model commonly used to simulate atmospheric and climate processes. We simulate a 10-dimensional system ($p = 10$) with the forcing constant set to 10 and generate 5,000 samples for training and evaluation.

**CF-Diamond.** To evaluate causal discovery on a structurally non-trivial topology that combines mediators, a collider, and self-causation, we adopt the diamond synthetic benchmark introduced by Kong et al. (2025). The system consists of four variables $\{x^{(1)}, x^{(2)}, x^{(3)}, x^{(4)}\}$ coupled through the directed causal structure $x^{(1)} \rightarrow x^{(2)}, x^{(1)} \rightarrow x^{(3)}, x^{(2)} \rightarrow x^{(4)}, x^{(3)} \rightarrow x^{(4)}$, with self-causal links $x^{(i)} \rightarrow x^{(i)}$, $\forall i \in \{1, \dots, 4\}$. The benchmark mixes lagged and instantaneous dependencies on the collider variable, providing a stringent test of mechanism modularity in the presence of overlapping ancestral paths. We use 4000 samples generated under the original simulation protocol with additive standard Gaussian noise.

**Nonlinear VAR (NL-VAR).** To test recovery under nonlinear dynamics with a random graph topology, we simulate a 10-dimensional first-order autoregressive process with quadratic interaction terms and a smooth saturating transition. Each variable receives three randomly selected non-self causal parents from a fixed sparse adjacency matrix, yielding a graph density of 0.4 when self-loops are included. We generate 5000 samples after discarding 500 burn-in steps.

***E. coli* dataset.** The *E. coli* dataset (Krieger & Gilpin, 2025) consists of in silico gene expression time series generated from transcriptional regulatory subnetworks derived from RegulonDB v6.7, where biologically grounded network topologies are coupled with simulated nonlinear regulatory dynamics to provide known causal ground-truth structures for evaluation.

**Yeast dataset.** The yeast dataset (Krieger & Gilpin, 2025) comprises simulated gene expression time series constructed from curated *Saccharomyces cerevisiae* transcriptional regulatory networks, combining experimentally derived regulatory interactions with mechanistic nonlinear dynamical models to yield benchmark causal graphs with known ground truth.

More details of the datasets, including data generation process, the governing equations, and simulation dynamics, are provided in Appendix A and B. Collectively, the benchmark suite spans chaotic, dissipative, and saturating nonlinear dynamics, deterministic and stochastic data-generating processes, chain, diamond, dense, and random graph topologies, and synthetic and biologically grounded regulatory networks. This

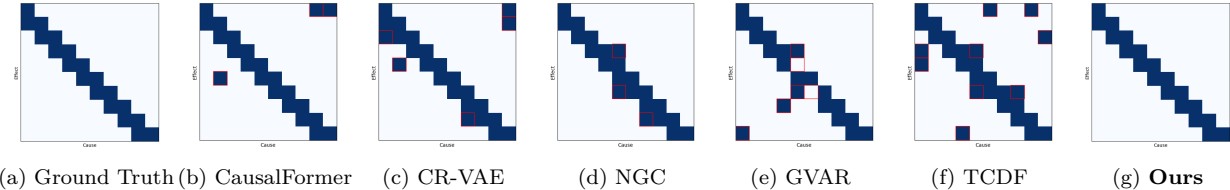

| (a) Ground Truth | (b) CausalFormer | (c) CR-VAE | (d) NGC | (e) GVAR | (f) TCDF | (g) **Ours** |

Figure 2: Visual representation of ground truth causal graph of Hénon (2a), retrieved adjacency matrices by the chosen baselines (2b-2f), and our proposed CauFR-TS (2g).

diversity ensures that the evaluation probes robustness across the principal dimensions along which causal discovery difficulty varies in practice.

## 5.2 Baselines

We compare our proposed CauFR-TS approach with five representative state-of-the-art methods for Granger Causal Discovery in multivariate time series. We specifically emphasize neural network-based approaches that infer causality via sparsification or interpretation of learned model parameters.

NGC (Tank et al., 2021) is the first neural network-based temporal causal discovery method which uses autoregressive neural networks with group sparsity on input connections. GVAR (Marcinkevičs & Vogt, 2021) uses self-explaining neural networks to augment neural Granger causality with explicit attribution mechanisms in order to improve interpretability of causal contributions. However, the underlying architecture still relies on shared latent representations and post-hoc explanation, failing to meet the ICM principles. CR-VAE (Li et al., 2023) integrates Granger causality into a recurrent variational autoencoder using a multi-head decoder and sparsity-inducing regularization. Causal structure is extracted from decoder weights, while the latent encoder is shared across variables, producing entangled representations. TCDF (Nauta et al., 2019) employs attention-based dilated temporal convolutions and infers causality from learned convolutional and attention weights. It represents convolutional neural approaches to Granger causal discovery based on parameter inspection under shared representations. Recently introduced CausalFormer (Kong et al., 2025) is a transformer-based temporal causal discovery model that learns predictive representations and recovers causal graphs through decomposition-based interpretation of attention and convolutional components.

All the aforementioned baselines apply sparsity or interpretability mechanisms, and infer causal links from globally shared representations. They implicitly assume that entangled latent embeddings preserve the conditional independence structure required for Granger causality. Our work formally examines and addresses the failure of this assumption by enforcing representation-level modularity through a factorized encoder-decoder design.

## 6 Results and Evaluation

The experimental evaluation assesses the proposed framework along two complementary dimensions. We first evaluate causal structure recovery from multivariate time series, which constitutes the primary objective of this work. We then assess predictive and generative performance to verify that enforcing representation-level modularity does not compromise forecasting accuracy. Causal discovery results are reported on synthetic and biologically grounded benchmarks with known ground truth, followed by forecasting results presented as a sanity check.

### 6.1 Causal Discovery Performance

Table 1 presents a quantitative comparison of causal discovery performance across the selected baseline methods and the proposed approach. Evaluation is conducted using three widely adopted metrics in the recent literature: Area under the receiver operating characteristic curve (AUROC) to assess ranking quality of inferred causal strengths; F1-score to measure binary edge detection accuracy after adaptive thresholding; Structural Hamming Distance (SHD) to quantify the absolute structural discrepancy between estimated and

ground-truth graphs. For each method, the causal matrix corresponding to the minimum convex loss is selected. CauFR-TS achieves perfect recovery on the Hénon and Lorenz-96 benchmarks across all three metrics, demonstrating the effectiveness of factorized representations on chaotic chain and dense lagged topologies. On NL-VAR, the proposed method attains the highest AUROC and F1-score, consistent with the lowest SHD, confirming that the architectural guarantees of variable separation extend to nonlinear regimes with random sparse topologies and saturating dynamics. On the CF-Diamond benchmark, CauFR-TS achieves perfect ranking but a marginally lower F1-score than CausalFormer and TCDF. This gap is attributable to the small-graph regime: with $D = 4$, the empirical distribution of decoder weight norms contains only 16 points, which limits the precision of the GMM-based threshold; the perfect AUROC nevertheless indicates that causal and non-causal links are correctly ranked. On the in silico *E. coli* and Yeast benchmarks, CauFR-TS yields the largest absolute improvements over the baselines across all three metrics, in line with the theoretical prediction that representation-level modularity is most beneficial when latent confounding compounds with measurement noise and overlapping regulatory pathways. The visual comparison of causal discovery from Hénon dataset has been given in Figure 2. Additional exploratory analyses of the decoder weight distributions, including posterior density visualizations and t-SNE visualizations of decoder weight norms, are provided in Appendix F.3.

**Remark 1 (Latent Deconfounding via Factorization)** *Let $X_t = \{x_{t-}^1, \ldots, x_{t-}^D\}$ be the multivariate history. In a shared-encoder framework, the latent state $z_t = E_\Phi(X_t)$ creates a path $x_{t-}^j \to z_t \to \hat{x}_t^i$ regardless of the causal graph structure $A_{ji}$. In contrast, CauFR-TS enforces a factorized mapping $z_t^i = E_\Phi^{(i)}(x_{t-}^i)$. Consequently, the conditional predictive distribution $P(x_t^i|X_t)$ satisfies:*

$$\frac{\partial \hat{x}_t^i}{\partial x_{t-}^j} \neq 0 \iff A_{ji} \neq 0$$

*This condition ensures that the adjacency matrix $A$ is the exclusive gateway for inter-variable information flow, structurally guaranteeing that non-zero edges in $A$ are necessary to minimize prediction error, thereby improving identifiability.*

To summarize, the observed performance gains of CauFR-TS can be attributed to its explicit enforcement of mechanism modularity at the representation level and the adaptive thresholding strategy applied on the decoder weight distribution. Unlike NGC, CausalFormer, and CR-VAE, which rely on shared latent encoders or post-hoc interpretation of entangled representations, CauFR-TS employs factorized, variable-specific encoders that prevent cross-channel information leakage during representation learning. This architectural constraint directly mitigates latent-induced confounding and restores the conditional independence assumptions underlying Granger causality. Furthermore, the proposed data-driven thresholding strategy based on the empirical distribution of decoder weights avoids the sensitivity and instability associated with manually tuned cutoffs, yielding consistent improvements.

## 6.2 Probabilistic Forecasting Performance

While the primary objective of this work is causal structure identification, CauFR-TS is trained as a generative forecaster and should therefore preserve predictive performance. We evaluate whether enforcing representation-level modularity and causal sparsity adversely affects forecasting accuracy. To this end, we compare one-step-ahead prediction performance against representative baselines using RMSE, a standard forecasting metric. The quantitative performance has been given in Table 2.

Across all datasets, CauFR-TS achieves predictive performance comparable to or better than shared-latent baselines. This indicates that enforcing factorized representations and causal gating does not compromise forecasting accuracy, despite significantly improving causal identifiability. Notably, CauFR-TS attains the lowest RMSE on every benchmark in Table 2, including the newly introduced CF-Diamond and NL-VAR systems, indicating that the predictive benefit of factorization is independent of the underlying graph topology and dynamical regime. These results support the view that factorized latent representation yields accurate temporal prediction, and therefore, a shared encoder can be safely eliminated in favour of causal modularity.

Table 1: Quantitative comparison of causal discovery using AUROC, F1-score, and SHD on the chosen datasets. The best performance has been highlighted in **bold**.

| Metrics | Datasets | CausalFormer (Kong et al., 2025) | CR-VAE (Li et al., 2023) | NGC (Tank et al., 2021) | GVAR (Marcinkevičs & Vogt, 2021) | TCDF (Nauta et al., 2019) | CauFR-TS (Proposed) |
|---|---|---|---|---|---|---|---|
| AUROC↑ | Hénon | 0.971 ($\pm$ 0.003) | 0.960 ($\pm$ 0.011) | 0.960 ($\pm$ 0.016) | 0.925 ($\pm$ 0.007) | 0.911 ($\pm$ 0.022) | **1.0 ($\pm$ 0.0)** |
| | Lorenz-96 | 0.975 ($\pm$ 0.006) | 0.954 ($\pm$ 0.013) | 0.980 ($\pm$ 0.008) | 0.862 ($\pm$ 0.031) | 0.871 ($\pm$ 0.023) | **1.0 ($\pm$ 0.0)** |
| | CF-Diamond | 0.945 ($\pm$ 0.042) | 0.896 ($\pm$ 0.066) | 0.861 ($\pm$ 0.008) | 0.799 ($\pm$ 0.017) | 0.813 ($\pm$ 0.016) | **1.0 ($\pm$ 0.0)** |
| | NL-VAR | 0.925 ($\pm$ 0.052) | 0.811 ($\pm$ 0.022) | 0.822 ($\pm$ 0.015) | 0.765 ($\pm$ 0.019) | 0.821 ($\pm$ 0.027) | **0.944 ($\pm$ 0.023)** |
| | E. coli | 0.785 ($\pm$ 0.003) | 0.766 ($\pm$ 0.005) | 0.760 ($\pm$ 0.012) | 0.654 ($\pm$ 0.032) | 0.596 ($\pm$ 0.041) | **0.994 ($\pm$ 0.002)** |
| | Yeast | 0.710 ($\pm$ 0.014) | 0.702 ($\pm$ 0.009) | 0.722 ($\pm$ 0.015) | 0.695 ($\pm$ 0.026) | 0.611 ($\pm$ 0.030) | **0.921 ($\pm$ 0.010)** |
| F1-Score↑ | Hénon | 0.926 ($\pm$ 0.005) | 0.879 ($\pm$ 0.030) | 0.946 ($\pm$ 0.012) | 0.866 ($\pm$ 0.014) | 0.820 ($\pm$ 0.025) | **1.0 ($\pm$ 0.0)** |
| | Lorenz-96 | 0.885 ($\pm$ 0.020) | 0.812 ($\pm$ 0.023) | 0.815 ($\pm$ 0.025) | 0.754 ($\pm$ 0.031) | 0.713 ($\pm$ 0.029) | **1.0 ($\pm$ 0.0)** |
| | CF-Diamond | **0.681 ($\pm$ 0.080)** | 0.632 ($\pm$ 0.090) | 0.556 ($\pm$ 0.192) | 0.615 ($\pm$ 0.051) | 0.681 ($\pm$ 0.090) | 0.659 ($\pm$ 0.012) |
| | NL-VAR | 0.802 ($\pm$ 0.012) | 0.715 ($\pm$ 0.032) | 0.722 ($\pm$ 0.009) | 0.756 ($\pm$ 0.041) | 0.695 ($\pm$ 0.011) | **0.853 ($\pm$ 0.031)** |
| | E. coli | 0.762 ($\pm$ 0.011) | 0.754 ($\pm$ 0.014) | 0.712 ($\pm$ 0.009) | 0.623 ($\pm$ 0.024) | 0.551 ($\pm$ 0.019) | **0.855 ($\pm$ 0.025)** |
| | Yeast | 0.689 ($\pm$ 0.021) | 0.675 ($\pm$ 0.015) | 0.701 ($\pm$ 0.009) | 0.678 ($\pm$ 0.010) | 0.584 ($\pm$ 0.020) | **0.819 ($\pm$ 0.018)** |
| SHD↓ | Hénon | 3 ($\pm$ 1) | 6 ($\pm$ 1) | 3 ($\pm$ 1) | 6 ($\pm$ 1) | 9 ($\pm$ 2) | **0 ($\pm$ 0)** |
| | Lorenz-96 | 5 ($\pm$ 2) | 4 ($\pm$ 1) | 4 ($\pm$ 2) | 6 ($\pm$ 2) | 8 ($\pm$ 3) | **0 ($\pm$ 0)** |
| | CF-Diamond | **4 ($\pm$ 1)** | 5 ($\pm$ 1) | 4 ($\pm$ 1) | 5 ($\pm$ 1) | 6 ($\pm$ 2) | 4 ($\pm$ 1) |
| | NL-VAR | 13 ($\pm$ 4) | 16 ($\pm$ 5) | 15 ($\pm$ 3) | 14 ($\pm$ 3) | 18 ($\pm$ 6) | **12 ($\pm$ 3)** |
| | E. coli | 10 ($\pm$ 2) | 12 ($\pm$ 3) | 11 ($\pm$ 2) | 10 ($\pm$ 2) | 17 ($\pm$ 3) | **6 ($\pm$ 3)** |
| | Yeast | 11 ($\pm$ 3) | 13 ($\pm$ 3) | 12 ($\pm$ 2) | 10 ($\pm$ 2) | 20 ($\pm$ 1) | **7 ($\pm$ 1)** |

Table 2: Quantitative comparison of one-step-ahead forecasting in terms of RMSE on the synthetic and in silico biological benchmarks. Lower values indicate better predictive accuracy.

| Methods | Hénon | Lorenz-96 | CF-Diamond | NL-VAR | E. coli | Yeast |
|---|---|---|---|---|---|---|
| NGC (Tank et al., 2021) | 0.233 | 0.384 | 0.565 | 0.241 | 0.668 | 0.689 |
| CR-VAE (Li et al., 2023) | 0.236 | 0.375 | 0.602 | 0.355 | 0.657 | 0.612 |
| **CauFR-TS (Proposed)** | 0.144 | 0.254 | 0.511 | 0.196 | 0.454 | 0.330 |

**Additional Information.** We provide additional visualization results and analyses on the convergence of the model and adaptive selection of threshold value in Appendix F, followed by the implementation setup in Appendix G.

## 6.3 Ablation Study

We conduct an ablation study to assess the contribution of the key architectural and algorithmic components of CauFR-TS. In particular, we examine the roles of encoder factorization and adaptive thresholding in achieving stable and identifiable causal discovery. Following the work by Li et al. (2023), we have chosen the fixed threshold value of 0 of the $\ell_2$-norm of decoder weights. Both the ablated variants are evaluated under the same experimental protocol and datasets as the full model.

The ablation results in Table 3 highlight the necessity of each component in the proposed framework. Replacing the factorized encoder with a shared encoder degrades causal graph recovery. This behaviour is consistent with the theoretical analysis, which shows that shared representations induce latent confounding and non-identifiability. Substituting the adaptive thresholding mechanism with a fixed cutoff leads to a significant drop in performance. Depending on the chosen threshold, the method either fails to remove noisy connections, resulting in a high false positive rate, or suppresses weak but genuine causal links, leading to false negatives. This instability highlights the limitation of manually specified thresholds and motivates data-driven causal link selection. The shared-encoder variant benefits from adaptive GMM-based thresholding. Although the latent representation remains entangled, the mixture model separates signal and noise in the decoder weight distribution with sufficient reliability. This allows partial recovery of the causal structure and yields performance comparable to strong baselines. Overall, the results show that causal identifiability in CauFR-TS emerges from the combined effect of representation-level modularity and adaptive thresholding.

Table 3: Quantitative evaluation of the impacts of encoder factorization and adaptive thresholding with their respective ablations on causal discovery performance on Hénon maps.

| Variant | Factorized Encoder | Adaptive Threshold | F1-score ↑ | SHD ↓ |
|---------|:---:|:---:|:---:|:---:|
| Shared Encoder | × | √ | 0.870 | 6 |
| Fixed Threshold | √ | × | 0.418 | 31 |
| **Full CauFR-TS** | √ | √ | **1.0** | **0** |

## 7 Conclusion

In this work, we investigated a fundamental limitation of the neural Granger causal discovery models for multivariate time series. Specifically, we showed that shared latent representations entangle distinct causal mechanisms and violate the conditional independence assumptions required for Granger causality, thereby leading to structurally non-identifiable causal graphs. To address these issues, we proposed CauFR-TS, a recurrent variational framework that enforces mechanism modularity through channel-wise encoders and mediates cross-variable dependencies exclusively via structured latent aggregation. In addition, we introduced an adaptive, data-driven thresholding strategy for causal link selection that avoids heuristic tuning and yields stable causal graphs. Empirical results on synthetic dynamical systems spanning chaotic, dissipative, and saturating nonlinear dynamics, as well as in silico biological benchmarks, demonstrate that CauFR-TS consistently matches or outperforms recent baselines in causal graph recovery. In addition, we demonstrate that the proposed factorized architecture preserves forecasting performance, confirming that causal identifiability can be achieved without sacrificing generative accuracy. Future work will extend this framework to time-varying and irregularly sampled settings and incorporate interventional or weakly supervised signals to further strengthen causal identifiability.

**Limitations.** The proposed framework is developed under the Granger causality paradigm and is therefore limited to predictive causal relationships inferred from temporal dependence. It does not model instantaneous effects or interventional causal semantics. The theoretical analysis assumes observational time series generated by stationary mechanisms. In addition, the use of dimension-wise encoders may introduce computational overhead in very high-dimensional settings.

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

# CauFR-TS: Causal Time-Series Identifiability via Factorized Representations
## (Appendix)

We have arranged the additional information and analysis in the appendix, organized under different sections and subsections. The arrangement of sections and their contents are as follows:

- Section A details the governing equations and data generation processes for the synthetic dynamical systems Hénon chaotic maps, Lorenz-96, CF-Diamond, and NL-VAR data.

- Section B describes the in silico biological benchmark datasets, including the underlying transcriptional regulatory networks and kinetic models.

- Section C provides the rigorous mathematical formulation of the proposed CauFR-TS architecture, emphasizing the factorized Transformer encoder and the group-sparse decoder under Sections C.1 and C.2 respectively.

- Section E defines the quantitative evaluation metrics used to assess causal graph recovery in this work.

- Section F presents extended empirical results, focusing on the convergence dynamics of the learned decoder weights during training, the stability and efficiency of the adaptive thresholding mechanism.

- Section G outlines the implementation details, including hardware specifications, hyperparameter settings, and optimization procedures.

## A   Synthetic Data Generation

Here we provide the governing equations and data generation procedures for the synthetic dynamical benchmarks employed in our evaluation.

**Hénon maps.** Following these equations, we have generated the Hénon chaotic maps (Kugiumtzis, 2013),

$$x_{t+1}^1 = 1.4 - (x_t^1)^2 + 0.3x_{t-1}^1, \tag{5}$$

$$x_{t+1}^d = 1.4 - (ex_t^{d-1} + (1-e)x_t^d)^2 + 0.3x_{t-1}^d, \tag{6}$$

where $d \in [2, M]$, $M$ is the dimensionality, $e = 0.3$ and $d = 10$. The true causal relations $x^d \to x^{d+1}$ and self-causal relations $x^d \to x^d$ in the corresponding adjacency matrix should be 1. The lag length has been taken as 2.

**Lorenz-96.** Similar to Hénon, we have used the following equations for simulating the M-dimensional Lorenz-96 maps (Lorenz, 1996),

$$\frac{dx_t^p}{dt} = (x_t^{p+1} - x_t^{p-2})x_t^{p-1} - x_t^p + F \tag{7}$$

where $x_t^{-1} = x_t^{M-1}, x_t^0 = x_t^M, x_t^{M+1} = x_t^1$, $p$ is the index variable and $F$ is the forcing constant that dictates the degree of nonlinearity and chaos in the series.

We generated around 5000 samples for both the datasets. For the Hénon and Lorenz-96 systems, we sample initial values from a typical Gaussian distribution and then infer the trajectories via transition functions.

**CF-Diamond.** We adopt the diamond synthetic benchmark proposed by Kong et al. (2025), which instantiates a four-variable causal system whose ground-truth graph follows a diamond topology. Let $\{x^{(1)}, x^{(2)}, x^{(3)}, x^{(4)}\}$ denote the variables. The benchmark specifies the directed structure: $x^{(1)} \to x^{(2)}$, $x^{(1)} \to x^{(3)}$, $x^{(2)} \to x^{(4)}$, $x^{(3)} \to x^{(4)}$, together with self-causal links $x^{(i)} \to x^{(i)}$, $\forall i \in \{1, \dots, 4\}$. The path $x^{(2)} \to x^{(4)}$ admits an instantaneous component. The data-generating process can be expressed compactly as,

$$x_t^{(p)} = f_p\left(\{x_{t-\ell}^{(q)} : (q, \ell) \in \mathrm{Pa}^\tau(p)\}\right) + \varepsilon_t^{(p)}, \quad \varepsilon_t^{(p)} \sim \mathcal{N}(0, 1) \tag{8}$$

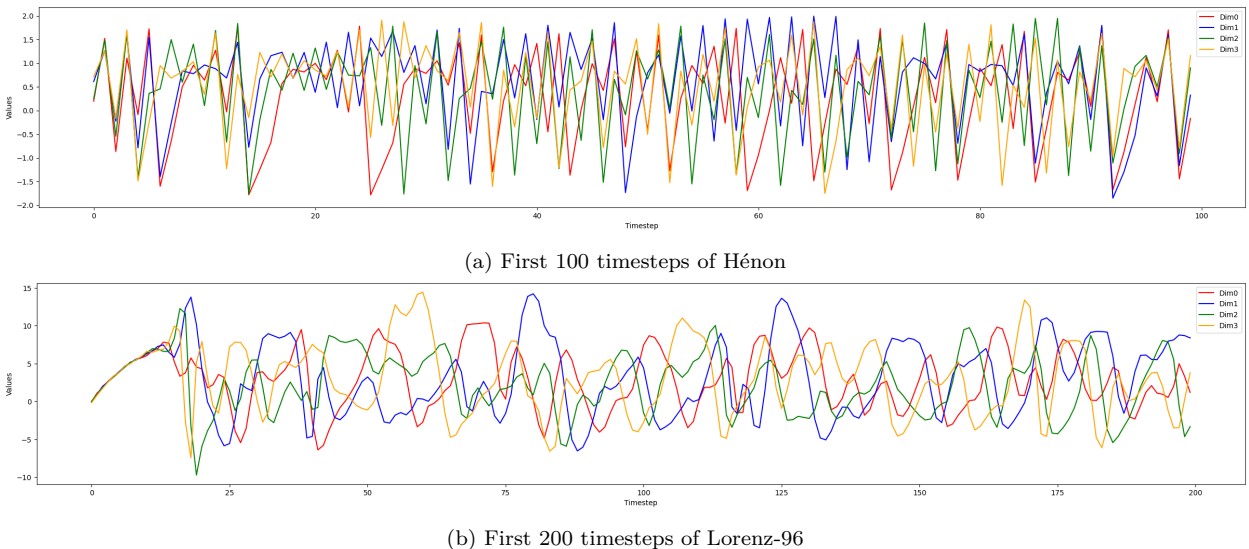

(a) First 100 timesteps of Hénon

(b) First 200 timesteps of Lorenz-96

Figure 3: Visualization of the first 4 dimensions of the generated Hénon and Lorenz-96 datasets

where $\mathrm{Pa}^\tau(p)$ denotes the set of (parent, lag) pairs of $x^{(p)}$ as specified by the diamond topology above, and $f_p$ is the variable-specific nonlinear transition function defined by the simulation protocol. The benchmark generates a single trajectory of 4000 time steps. The associated ground-truth adjacency matrix $A \in \{0,1\}^{4\times4}$ has unit entries at the positions $(2,1),(3,1),(4,2),(4,3)$, along with the principal diagonal entries denoting self-causations and all remaining entries equal to zero. The diamond benchmark complements the chain topology of Hénon and the dense topology of Lorenz-96 by introducing mediators, a collider, and self-causation simultaneously, thereby probing the recovery of mixed-lag dependencies that include both lagged and instantaneous components.

**NL-VAR.** To evaluate causal discovery under nonlinear dynamics with a random graph topology, we simulate a $D$-dimensional first-order autoregressive process augmented with quadratic parent interactions and a smooth saturating transition. Let $\mathrm{Pa}(i) \subset \{1,\ldots,D\} \setminus \{i\}$ denote the set of non-self causal parents of variable $i$, drawn from a fixed sparse adjacency matrix in which each variable receives exactly three non-self parents in addition to a self-loop. For each variable $i \in \{1,\ldots,D\}$, the latent pre-activation is defined as,

$$\tilde{x}_t^{(i)} = a_i\, x_{t-1}^{(i)} + \sum_{j\in\mathrm{Pa}(i)} \left[ b_{ij}\, x_{t-1}^{(j)} + c_{ij} \left(x_{t-1}^{(j)}\right)^2 \right] \tag{9}$$

The observed value is generated by applying a smooth nonlinear damping followed by additive noise:

$$x_t^{(i)} = \frac{\tilde{x}_t^{(i)}}{1 + 0.1\left(\tilde{x}_t^{(i)}\right)^2} + \varepsilon_t^{(i)}, \qquad \varepsilon_t^{(i)} \sim \sigma\, \mathcal{U}(0,1), \tag{10}$$

where $\sigma = 0.3$ controls the noise scale. The transition coefficients are sampled as $a_i \sim \mathcal{U}(0.2, 0.4)$, $b_{ij} = s_{ij}\beta_{ij}$ with $\beta_{ij} \sim \mathcal{U}(1.2, 1.5)$, and $c_{ij} = s_{ij}\gamma_{ij}$ with $\gamma_{ij} \sim \mathcal{U}(0.1, 0.2)$, where the sign $s_{ij} \in \{+1, -1\}$ is drawn independently for each parent edge with $P(s_{ij} = +1) = 0.9$. Sharing the sign between $b_{ij}$ and $c_{ij}$ ensures that the linear and quadratic contributions of each parent reinforce rather than cancel, which preserves the detectability of the causal effect under the smooth damping. The damping function $g(u) = \frac{u}{1+0.1u^2}$ is approximately linear near the origin and bounds large excursions, preventing the unbounded growth that would otherwise arise from the quadratic terms while avoiding the hard saturation of tanh.

We simulate with $D = 10$ over 5000 time steps after discarding 500 burn-in steps initialized from $\mathcal{N}(0, 0.25\mathbf{I})$. The graph is held fixed across runs to ensure reproducibility, and the ground-truth adjacency matrix $A \in \{0,1\}^{D\times D}$ has $A_{ii} = 1$ for all $i$ and $A_{ij} = 1$ if $j \in \mathrm{Pa}(i)$, yielding a total graph density of 0.4.

The generated data is then divided into several chunks of consecutive timesteps. Each of the chunks is considered as a single instance. We divide the set of instances into training and validation sets in a ratio of 85:15. For all the experiments, we have taken the chunk size $X_{1:T} = 50$.

## B   Details about in silico Biological Benchmark Datasets

*E. coli* **Dataset.** To evaluate causal discovery in a biologically grounded setting, we utilize the *E. coli* benchmark dataset introduced by Krieger & Gilpin (2025). This dataset consists of gene expression time series generated from transcriptional regulatory subnetworks subsampled from the E. coli interactome (RegulonDB v6.7). The underlying dynamics are simulated using a stochastic kinetic model that explicitly captures thermodynamic noise and regulatory kinetics. By retaining the topology of real biological subnetworks, such as specific degree distributions and motif frequencies, this dataset provides a realistic testbed for identifying causal links from sparse, noisy, and non-linear biological signals. For our experiments, we utilize the 10-node subsystem variant to evaluate the recovery of ground-truth transcriptional interactions.

**Yeast Dataset.** We further evaluate our method on the Yeast dataset (Krieger & Gilpin, 2025), which comprises simulated gene expression trajectories derived from *Saccharomyces cerevisiae*. Similar to the E. coli benchmark, the ground-truth causal graphs are constructed by integrating genome-wide transcription factor binding evidence with curated interactions. The dynamics are governed by a mechanistic model of coupled mRNA-protein interactions that incorporates key non-linear regulatory features, such as dimerization, phosphorylation-dependent activation, and autoregulatory feedback loops. These mechanisms induce complex dynamical behaviours, such as multistability and oscillations, making it an ideal dataset for validating our experiments on establishing causal identifiability.

## C   Details about the Encoder and Decoder

In this section, we provide the formal mathematical formulation of the channel-wise encoders and decoders introduced in Section 4.1. We employ a Transformer-based architecture for the encoder to capture long-range temporal dependencies within each univariate series and a structured Multi-Layer Perceptron (MLP) for the decoder to model cross-variable causal interactions.

### C.1   Factorized Transformer Encoder

For a $D$-dimensional input, our proposed CauFR-TS utilizes $D$ independent encoders. The input the $d$-th encoder head is the lagged history window $\mathbf{x}_{t-\tau:t-1}^{(d)} \in \mathbb{R}^{\tau \times 1}$.

**Input Embedding and Positional Encoding.** The scalar input at each time step $t \in \{1, \ldots, \tau\}$ is first projected into a higher-dimensional feature space $d_{model}$ using a linear layer, followed by the addition of sinusoidal positional encodings $PE$ to retain temporal order information:

$$\mathbf{h}_t^{(d)} = \text{Linear}(x_{T-\tau:t-1}^{(d)}) + PE(t)$$

**Self-Attention Mechanism.** The embedding is processed through $L$ layers of Transformer blocks. Each block $k$ computes the self-attention mechanism on the channel-specific history. Let $H^{(d,k-1)} \in \mathbb{R}^{\tau \times d_{model}}$ be the input to layer $k$. The query ($Q$), key ($K$), and value ($V$) matrices are computed as:

$$Q = H^{(d,k-1)}W_Q^{(d)}, \quad K = H^{(d,k-1)}W_K^{(d)}, \quad V = H^{(d,k-1)}W_V^{(d)}$$

The attention output is, therefore, derived via the scaled dot-product attention:

$$\text{Attention}(Q, K, V) = \text{softmax}\left(\frac{QK^T}{\sqrt{d_k}}\right)V$$

This mechanism allows the encoder $E_\Phi^{(d)}$ to dynamically weigh the historical time steps that are most relevant for the current state of variable $d$, without accessing information from other variables.

**Variational Reparameterization.** To enable the generative capability and robust latent representation required by the ELBO objective (Equation 2), the final hidden state of the Transformer is mapped to the parameters of the variational posterior $q_\phi(z^{(d)}|x^{(d)})$. Let $\mathbf{h}_{final}^{(d)}$ be the last time step output or a pooled representation, then we can represent the steps as:

$$\boldsymbol{\mu}^{(d)} = \text{Linear}_\mu(\mathbf{h}_{final}^{(d)}), \quad \log \boldsymbol{\sigma}^{2(d)} = \text{Linear}_\sigma(\mathbf{h}_{final}^{(d)})$$

The latent code $z^{(d)} \in \mathbb{R}^H$ is then obtained via the reparameterization trick:

$$z^{(d)} = \boldsymbol{\mu}^{(d)} + \boldsymbol{\epsilon} \odot \boldsymbol{\sigma}^{(d)}, \quad \boldsymbol{\epsilon} \sim \mathcal{N}(0, I)$$

This ensures that the latent subspace for dimension $d$ contains stochastic information derived exclusively from the historical observations $x_{t-\tau:t-1}^{(d)}$. The global latent representation $\mathbf{z} \in \mathbb{R}^{D \times H}$ is formed by concatenating the independent latent vectors $\mathbf{z} = [z^{(1)}, z^{(2)}, \ldots, z^{(D)}]$.

## C.2 Group-Sparse Decoder

The prediction for the target variable $x_t^{(d)}$ is generated by a dedicated decoder $D_\Theta^{(d)}$. To strictly enforce the Granger causal definition, the decoder models the mapping from the joint history $\mathbf{z}$ to the target distribution.

**Linear Projection and Sparsification.** The core of the causal discovery mechanism lies in the decoder heads. The input $\mathbf{z}$ is formed as a vector of size $D \times H$. The decoder applies a weight matrix $\mathbf{W}^{(d)} \in \mathbb{R}^{H_{out} \times (D \times H)}$:

$$\mathbf{u}^{(d)} = \mathbf{W}^{(d)}.\mathbf{z} + \mathbf{b}^{(d)}$$

The weight matrix $\mathbf{W}^{(d)}$ is structurally partitioned into $D$ groups, where the $j$-th group $\mathbf{W}_{(j)}^{(i)} \in \mathbb{R}^{H_{out} \times H}$ corresponds to the weights multiplying the latent sub-vector $z^{(j)}$. The Group Lasso penalty (third component in Equation 2) operates on these blocks:

$$\mathcal{L}_{GL} = \lambda \sum_{i=1}^{D} \sum_{j=1}^{D} \|W_{(j)}^i\|_2$$

If $\|W_{(j)}^{(i)}\|_2 \to 0$, the decoder $D^{(i)}$ effectively ignores the information from variable $j$, implying $x^{(j)}$ does not Granger-cause $x^{(i)}$.

**Probabilistic Output.** The output of the linear projection maps to the parameters of the predictive distribution for the target variable:

$$\hat{x}_t^{(d)} \sim \mathcal{N}(\mu_{dec}^{(d)}, \sigma_{dec}^{2(d)}) = D_\Theta^{(d)}(\mathbf{z})$$

This formulation of multi-head decoder ensures that the only path for information to flow from variable $j$ to variable $i$ is through the specific weight group $W_{(j)}^i$. It makes the causal structure explicitly identifiable via the adaptive thresholding mechanism described in Algorithm 2.

# D Theoretical Proofs

In this section, we provide formal derivations for the propositions stated in Section 3 and Section 6. We rigorously show that shared latent encoders introduce irreducible confounding that violates the structural conditions required for Granger causal discovery and that our proposed CauFR-TS restores the necessary conditional independence.

**Assumptions.** The following proofs operate under these assumptions:

(A1) **Differentiability:** The shared encoder $f_\theta$ is differentiable almost everywhere, which holds for standard activations such as ReLU, tanh, and GELU;

(A2) **Non-degenerate parameterization:** The weight matrices of $f_\theta$ are initialized with full rank and are not explicitly constrained toward block-diagonal or sparse structure;

(A3) **Predictive training objective:** $f_\theta$ is optimized by minimizing a global reconstruction or forecasting loss over all $D$ variables jointly, without any variable-wise independence regularization;

(A4) **Correlated inputs:** The input variables $x^{(1)}, \ldots, x^{(D)}$ are not mutually statistically independent, which is the standard assumption in any meaningful causal discovery setting.

Under these conditions, Propositions 1 and 2 hold in the general case. Both proofs rely on a shared structural property of the encoder Jacobian, which we first establish as an independent result. The following Lemma 1 formalizes the claim that a shared encoder satisfying (A1) and (A2) yields a generically dense Jacobian, thereby precluding the block-diagonal structure required for variable-wise separation. We present this result separately, as it forms the common basis for both propositions.

**Lemma 1 (Generic Density of the Shared Encoder Jacobian)** *Let $f_\theta : \mathbb{R}^{\tau \times D} \to \mathbb{R}^H$ be a shared encoder satisfying* (A1) *and* (A2). *Then the Jacobian $J = \frac{\partial \mathbf{z}_t}{\partial \mathbf{X}_{t-}}$ contains no structurally zero entries: for Lebesgue-almost every parameter-input pair $(\theta, \mathbf{X}_{t-})$, every entry $J_{m,k}$ is non-zero. Equivalently, the set*

$$\mathcal{S} = \left\{ (\theta, \mathbf{X}_{t-}) : J_{m,k}(\theta, \mathbf{X}_{t-}) = 0 \text{ for some } m, k \right\}$$

*has Lebesgue measure zero.*

**Proof of Lemma 1.**

By the chain rule, the Jacobian of the $L$-layer encoder decomposes as:

$$J = \mathbf{D}^{(L)} \mathbf{W}^{(L)} \mathbf{D}^{(L-1)} \mathbf{W}^{(L-1)} \cdots \mathbf{D}^{(1)} \mathbf{W}^{(1)}$$

where $\mathbf{D}^{(l)} = \mathrm{diag}\big(\sigma'(a^{(l)})\big)$ is the diagonal matrix of activation derivatives evaluated at the pre-activations $a^{(l)} = \mathbf{W}^{(l)} h^{(l-1)} + b^{(l)}$.

**(i) Non-degeneracy of activation derivatives.** Under (A1), the derivative $\sigma'$ vanishes only on a measure-zero subset of $\mathbb{R}$: for ReLU this is the singleton $\{0\}$; for tanh and GELU, $\sigma' > 0$ everywhere. Each pre-activation $a_m^{(l)}$ is a non-constant affine function of $(\theta, \mathbf{X}_{t-})$ under (A2). Therefore, the set of $(\theta, \mathbf{X}_{t-})$ for which any diagonal entry $\sigma'(a_m^{(l)}) = 0$ lies in a proper algebraic subvariety of the parameter-input space and has Lebesgue measure zero. It follows that, for almost every $(\theta, \mathbf{X}_{t-})$, each $\mathbf{D}^{(l)}$ is a diagonal matrix with all non-zero diagonal entries.

**(ii) Non-vanishing of the product.** Under (A2), each $\mathbf{W}^{(l)}$ has full rank. Each entry $J_{m,k}$ of the product $J$ is a polynomial function of the entries of $\{\mathbf{W}^{(l)}, b^{(l)}, \mathbf{X}_{t-}\}$ (for piecewise-linear activations such as ReLU, this holds within each linear activation region; for smooth activations such as tanh and GELU, $J_{m,k}$ is real-analytic, and the following argument applies identically). This polynomial is not identically zero: at any initialization satisfying (A2) with all activation derivatives positive, $J$ is a product of full-rank dense matrices and invertible diagonal matrices, which yields a matrix with all non-zero entries. Since a non-identically-zero polynomial vanishes only on a set of Lebesgue measure zero (Schwartz, 1980), each $J_{m,k} \neq 0$ almost everywhere. As the overall zero set $\mathcal{S}$ is a finite union of such measure-zero sets (one per entry), $\mathcal{S}$ itself has measure zero.

## D.1 Proof of Proposition 1

Consider the shared encoder mapping $f_\theta : \mathbb{R}^{\tau \times D} \to \mathbb{R}^H$. Let $\mathbf{X}_{t-}$ be the input history window. The latent vector is given by $\mathbf{z}_t = f_\theta(\mathbf{X}_{t-})$. For the representation to admit a variable-wise decomposition $\mathbf{z}_t = (z_t^{(1)}, \ldots, z_t^{(D)})$ such that each component $z_t^{(p)}$ depends exclusively on $x_{t-}^{(p)}$, the Jacobian matrix $J = \frac{\partial \mathbf{z}_t}{\partial \mathbf{X}_{t-}}$ must exhibit a block-diagonal or permutation-equivalent structure:

$$\frac{\partial z^{(p)}}{\partial x^{(q)}} = 0 \quad \text{for all } p \neq q$$

**Non-zero Jacobian structure.** By Lemma 1, under assumptions (A1) and (A2), every entry of $J$ is non-zero for Lebesgue-almost every $(\theta, \mathbf{X}_{t-})$. In particular, the off-diagonal blocks $\frac{\partial z^{(p)}}{\partial x^{(q)}}$ are generically non-zero for all $p \neq q$, which directly contradicts the block-diagonal requirement above.

**Non-decomposability.** It follows that no partition of $\mathbf{z}_t$ into $D$ variable-specific sub-vectors exists such that each sub-vector depends exclusively on a single input channel. Since each latent coordinate $z_{t,m}$ is a function of the full multivariate history, we can write:

$$z_{t,m} = g_m(x_{t-}^{(1)}, x_{t-}^{(2)}, \ldots, x_{t-}^{(D)})$$

where $g_m$ depends non-trivially on multiple inputs. Under (A4), which ensures the input variables carry non-redundant information, the mutual information $I(z_{t,m}; x_{t-}^{(q)}) > 0$ for all $q \in \{1, \ldots, D\}$. Therefore, a valid causal decomposition requiring sub-vectors $z^{(p)}$ such that $I(z^{(p)}; x_{t-}^{(q)}) = 0$ for all $q \neq p$ cannot be achieved. It clearly establishes that the shared encoder produces a non-separable, entangled representation.

### D.2 Proof of Proposition 2

**Formulation of Causal Estimator.** Consider the shared encoder mapping $f_\theta : \mathbb{R}^{\tau \times D} \to \mathbb{R}^H$. Let $\mathbf{X}_{t-}$ be the input history window. The latent vector is given by $\mathbf{z}_t = f_\theta(\mathbf{X}_{t-})$. The prediction for the $i$-th target variable, $\hat{x}_t^{(i)}$, is generated by a decoder function $D_\Theta^{(i)}$ acting on this latent state:

$$\hat{x}_t^{(i)} = D_\Theta^{(i)}(\mathbf{z}_t) = \mathbb{E}[x_t^{(i)} \mid \mathbf{z}_t]$$

For a causal link $j \to i$ to be identified as non-existent, the model must exhibit invariance to the history of $j$ when predicting $i$. Mathematically, this requires the gradient of the prediction with respect to the input $x_{t-}^{(j)}$ to be zero:

$$\frac{\partial \hat{x}_t^{(i)}}{\partial x_{t-}^{(j)}} = 0 \quad \text{if } j \notin PA(x^{(i)})$$

**Step 1: Gradient decomposition.** By the multivariate chain rule, the total sensitivity of the prediction $\hat{x}_t^{(i)}$ to the input history $x_{t-}^{(j)}$ decomposes as:

$$\frac{\partial \hat{x}_t^{(i)}}{\partial x_{t-}^{(j)}} = \sum_{h=1}^{H} \underbrace{\frac{\partial \hat{x}_t^{(i)}}{\partial z_{t,h}}}_{\text{Decoder}} \cdot \underbrace{\frac{\partial z_{t,h}}{\partial x_{t-}^{(j)}}}_{\text{Encoder}} = \mathbf{d}_i^\top \mathbf{e}_j \tag{11}$$

where we define the decoder sensitivity vector $\mathbf{d}_i = \left(\frac{\partial \hat{x}_t^{(i)}}{\partial z_{t,1}}, \ldots, \frac{\partial \hat{x}_t^{(i)}}{\partial z_{t,H}}\right)^\top \in \mathbb{R}^H$ and the encoder sensitivity vector $\mathbf{e}_j = \left(\frac{\partial z_{t,1}}{\partial x_{t-}^{(j)}}, \ldots, \frac{\partial z_{t,H}}{\partial x_{t-}^{(j)}}\right)^\top \in \mathbb{R}^H$.

**Step 2: Analysis of the encoder term.** From Proposition 1 (via Lemma 1), for generic $(\theta, \mathbf{X}_{t-})$, every entry of $\mathbf{e}_j$ is non-zero for all $j \in \{1, \ldots, D\}$. That is, a perturbation in the input $x_{t-}^{(j)}$ propagates to every dimension of $\mathbf{z}_t$.

**Step 3: Analysis of the decoder term via predictive necessity.** Suppose $x^{(i)}$ has at least one true causal parent $x^{(k)}$, i.e., $k \in PA(x^{(i)})$, and consider a non-parent $j \notin PA(x^{(i)})$. Under (A3), the decoder $D_\Theta^{(i)}$ is trained to minimize the prediction error for $x_t^{(i)}$. To achieve non-trivial predictive accuracy, the decoder must extract information about the true parent $x_{t-}^{(k)}$ from $\mathbf{z}_t$. The only pathway from $x_{t-}^{(k)}$ to $\hat{x}_t^{(i)}$ passes through $\mathbf{z}_t$, which requires $\mathbf{d}_i^\top \mathbf{e}_k \neq 0$. In particular, $\mathbf{d}_i \neq \mathbf{0}$.

**Step 4: Non-vanishing of the spurious gradient.** For correct causal identification, we would need $\mathbf{d}_i^\top \mathbf{e}_j = 0$ for every non-parent $j \notin PA(x^{(i)})$, i.e., $\mathbf{d}_i$ must lie in the orthogonal complement of $\text{span}\{\mathbf{e}_j : j \notin PA(x^{(i)})\}$. However, under the shared encoder, the encoder sensitivity vectors $\{\mathbf{e}_1, \ldots, \mathbf{e}_D\}$ are generically linearly independent (since the columns of the Jacobian of a generic smooth map are linearly independent

when $H \geq D$, which is the standard over-parameterized regime). The parent and non-parent encoder vectors therefore span overlapping, generically non-aligned directions in $\mathbb{R}^H$, and there is no architectural mechanism that constrains $\mathbf{d}_i$ to be orthogonal to all non-parent directions while remaining non-orthogonal to parent directions.

Formally, $\mathbf{d}_i^\top \mathbf{e}_j$ is a polynomial function of $(\theta, \mathbf{X}_{t-})$ (with the same piecewise or analytic qualification as in Lemma 1). This polynomial is not identically zero: at any parameter configuration where the decoder assigns uniform positive sensitivity across all latent dimensions (i.e., $\mathbf{d}_i$ has all positive entries) and all encoder sensitivity entries are positive (which holds at generic initializations satisfying (A1)–(A2)), the inner product $\mathbf{d}_i^\top \mathbf{e}_j > 0$. By the same argument as in Lemma 1, this quantity vanishes only on a set of Lebesgue measure zero.

Therefore, for generic parameters:

$$\frac{\partial \hat{x}_t^{(i)}}{\partial x_{t-}^{(j)}} = \mathbf{d}_i^\top \mathbf{e}_j \neq 0 \quad \text{even when } j \notin PA(x^{(i)})$$

The shared encoder creates an irreducible confounding path through the latent space. The decoder cannot selectively access information about a true parent $x^{(k)}$ without simultaneously inheriting sensitivity to the non-parent $x^{(j)}$, because both contribute to the same latent dimensions. This violates the core condition of Granger Non-Causality, which requires that $P(x_t^{(i)} \mid \mathbf{X}_{t-}) = P(x_t^{(i)} \mid \mathbf{X}_{t-}^{j \notin PA(x^{(i)})})$. Thus, the causal graph derived from such a model is structurally non-identifiable.

**Remark (Identity Map and Scope of Proposition 2).** A natural question is whether a shared encoder that learns the identity map, i.e., one where $z_t \approx X_{t-}$, would be immune to the confounding described above. This is a degenerate edge case that requires the latent dimensionality $H$ to match the input dimensionality $\tau \times D$, and even then, variable-wise separation is not guaranteed: the identity map preserves all cross-variable correlations in the input and does not induce a block-diagonal Jacobian unless the input variables are already statistically independent. More critically, there is no mechanism in a standard predictive training objective that would drive a shared encoder toward such a solution. The optimizer exploits all available correlations to minimize forecasting error, and the resulting Jacobian is dense almost everywhere, as established by Lemma 1. Proposition 2 therefore holds in the general case of shared encoders trained under predictive objectives, independently of whether a block-diagonal solution exists in principle.

### D.3 Justification on Remark 1

**Formulation of Factorized Estimator.** Unlike the shared formulation in Proposition 2, CauFR-TS processes each input dimension $k \in \{1, \ldots, D\}$ via an independent encoder $E_\phi^{(k)}$. The latent representation for variable $k$ is conditioned exclusively on its own history:

$$z_t^{(k)} = E_\Phi^{(k)}(x_{t-}^{(k)})$$

The global latent state $z_t$ is constructed via concatenating all the embeddings from the Encoder heads as $z_t = [z_t^{(1)}, z_t^{(2)}, \ldots, z_t^{(D)}]$. The prediction for a target variable $i$ is generated by a channel-specific decoder $D_\theta^{(i)}$, which takes the global state $z_t$ as input:

$$\hat{x}_t^{(i)} = D_\Theta^{(i)}(z_t) = D_\theta^{(i)}([z_t^{(1)}, \ldots, z_t^{(D)}])$$

**Analysis of the Jacobian Structure.** The global Jacobian $\frac{\partial z_t}{\partial X_{t-}}$ describes how every latent component changes with every input. Because the encoder $E_\Phi^{(k)}$ does not receive inputs from any variable $j \neq k$, the below expression holds true:

$$\frac{\partial z_t^{(k)}}{\partial x_{t-}^{(j)}} = \begin{cases} J_k & \text{if } k = j \\ \mathbf{0} & \text{if } k \neq j \end{cases} \tag{12}$$

This makes the Jacobian for the encoder strictly block-diagonal, fundamentally distinct from the dense Jacobian of the shared encoder derived in Lemma 1.

**Collapsing Spurious Paths.** To determine if source $j$ Granger-causes target $i$, we evaluate the total sensitivity of the prediction $\hat{x}_t^{(i)}$ to the input history $x_{t-}^{(j)}$:

$$\frac{\partial \hat{x}_t^{(i)}}{\partial x_{t-}^{(j)}} = \sum_{k=1}^{D} \underbrace{\left(\frac{\partial \hat{x}_t^{(i)}}{\partial z_t^{(k)}}\right)}_{\text{Path via } z^{(k)}} \cdot \underbrace{\left(\frac{\partial z_t^{(k)}}{\partial x_{t-}^{(j)}}\right)}_{\text{Sensitivity of } z^{(k)}} \tag{13}$$

For any path where $k \neq j$, the term $\frac{\partial z_t^{(k)}}{\partial x_{t-}^{(j)}}$ is zero. This means the history of variable $j$ cannot mix into the latent code of variable $k$. Therefore, the only non-zero term is the term where $k = j$. Substituting the block-diagonal property (Equation 12) into Equation 13, it collapses into a single causal path:

$$\frac{\partial \hat{x}_t^{(i)}}{\partial x_{t-}^{(j)}} = \frac{\partial \hat{x}_t^{(i)}}{\partial z_t^{(j)}} \cdot \frac{\partial z_t^{(j)}}{\partial x_{t-}^{(j)}} \tag{14}$$

This shows that the influence of $x^{(j)}$ on $x^{(i)}$ is mediated exclusively by the specific latent vector $z^{(j)}$. It effectively blocks all backdoor paths through other latent dimensions that existed in the shared architecture.

**Role of the Group-Sparse Decoder.** The term $\frac{\partial \hat{x}_t^{(i)}}{\partial z_t^{(j)}}$ represents the sensitivity of the $i$-th decoder to the $j$-th latent code. In our proposed CauFR-TS, this sensitivity is parameterized linearly by the weight group $\mathbf{W}_{(j)}^i$:

$$\frac{\partial \hat{x}_t^{(i)}}{\partial z_t^{(j)}} \propto \mathbf{W}_{(j)}^i$$

The objective function (Equation 2) applies a Group Lasso penalty to drive $||\mathbf{W}_{(j)}^i||_2 \to 0$ for non-causal pairs. During the optimization, if $\mathbf{W}_{(j)}^i$ becomes 0, then $\frac{\partial \hat{x}_t^{(i)}}{\partial z_t^{(j)}} = 0$. Therefore, this forces the total gradient $\frac{\partial \hat{x}_t^{(i)}}{\partial x_{t-}^{(j)}} = 0$ in Equation 14. If $\mathbf{W}_{(j)}^i \neq 0$, the gradient remains non-zero.

To summarize, the factorization ensures that the condition $\frac{\partial \hat{x}_t^{(i)}}{\partial x_{t-}^{(j)}} \neq 0$ is mathematically equivalent to $\mathbf{W}_{(j)}^i \neq 0$. This guarantees that the inferred adjacency matrix $\hat{A}$ accurately reflects the true conditional dependence structure of the data, thereby fulfilling the requirements of structural identifiability.

## E  Evaluation Metrics

Here we provide a brief description and equations of the quantitative evaluation metrics we have used in our work.

**Area under Receiver Operating Characteristic curve (AUROC).** The Receiver Operating Characteristic (ROC) curve is a graphical representation that plots the True Positive Rate (TPR) against the False Positive Rate (FPR) at different threshold values. The AUROC represents the area under this curve, providing a single scalar value that summarizes the model's performance. The AUC ranges from 0 to 1, where, an AUC of 0.5 indicates no discriminative ability and an AUC of 1.0 indicates perfect discrimination between classes. It can be mathematically represented as shown in Equation 15.

$$\text{AUROC} = \sum_{i=1}^{N} (\text{FPR}_i - \text{FPR}_{i-1}).\text{TPR} \tag{15}$$

where TPR and FPR stand for True Positive Rate and False Positive Rate, respectively, and they can be given as follows:

$$\text{TPR} = \frac{\#\text{True Positives}}{\#\text{True Positives} + \#\text{False Negatives}}$$

$$\text{FPR} = \frac{\#\text{False Positives}}{\#\text{False Positives} + \#\text{True Negatives}}$$

We use the AUROC score as a quantitative indicator while assessing the calculated causal adjacency matrices by comparing them to the ground truth. The numerical results have been given in the main manuscript.

**F1-score.** While AUROC assesses the global ranking quality of the learned interaction strengths, it does not evaluate the final recovered causal structure requiring a discrete decision boundary. We employ the F1-score in order to evaluate the quality of the binary causal graph $\hat{A}$. This metric is particularly critical in our setting to validate the effectiveness of the proposed adaptive thresholding strategy described in Section 4.3. Given the ground truth binary adjacency matrix $A \in \{0,1\}^{D \times D}$ and the estimated binary matrix $\hat{A} \in \{0,1\}^{D \times D}$ obtained after thresholding the decoder weights, the F1-score is calculated as:

$$\text{F1} = 2 \cdot \frac{\text{Precision} \cdot \text{Recall}}{\text{Precision} + \text{Recall}} \tag{16}$$

where Precision $= \frac{\text{TP}}{\text{TP}+\text{FP}}$ and Recall $= \frac{\text{TP}}{\text{TP}+\text{FN}}$. We define the constituent counts as:

- True Positives (TP): The number of correctly identified causal links $(i,j)$ such that $\hat{A}_{ij} = 1$ and $A_{ij} = 1$.

- False Positives (FP): The number of identified spurious links $(i,j)$ such that $\hat{A}_{ij} = 1$ and $A_{ij} = 0$.

- False Negatives (FN): The number of missed causal links $(i,j)$ such that $\hat{A}_{ij} = 0$ and $A_{ij} = 1$.

An F1-score of 1.0 indicates perfect reconstruction of the causal skeleton, whereas lower scores reflect a trade-off between missing true edges and hallucinating spurious ones.

**Structural Hamming Distance (SHD).** To quantify the absolute topological discrepancy between the estimated graph and the ground truth, we utilize the Structural Hamming Distance (SHD). Unlike the F1-score, SHD provides a direct count of the structural errors, specifically the number of edge insertions, deletions, or reversals required to transform the estimated graph $\hat{A}$ into the true graph $A$. SHD is formally defined as the $\ell_0$ norm of the difference between the adjacency matrices of the ground truth and the retrieved causal graph:

$$\text{SHD}(A, \hat{A}) = \sum_{i=1}^{D} \sum_{j=1}^{D} \mathbb{I}(\hat{A}_{ij} \neq A_{ij}) \tag{17}$$

where $\mathbb{I}(\cdot)$ is the indicator function. A value of SHD = 0 implies that the recovered graph is isomorphic to the ground truth. This metric is particularly sensitive to the density of the graph. Therefore, we consider it alongside AUROC and F1-score to provide a comprehensive view of identifiability.

## F  Additional Results and Discussions

### F.1  Convergence Analysis and Plots

To empirically validate the optimization dynamics of CauFR-TS, we analyze the evolution of the learned decoder weights and the stability of the proposed adaptive thresholding mechanism throughout the training process. Figures 4, 5, and 6 illustrate the trajectory of the raw Granger Causal (GC) values, specifically , the $\ell_2$-norms of the decoder weight groups $||W_{(j)}^i||_2$, across the dimensions of all six benchmark datasets over the training epochs.

**Separation of Causal and Non-Causal Mechanisms.** The plots in Figures 4 and 5 demonstrate the creation of a distinct separation between the weights associated with true causal parents and those associated with non-causal variables towards the end. The weights corresponding to ground-truth causal links showed a rapid ascent at the beginning, subsequently stabilizing at significant magnitudes. This confirms that the factorized encoder-decoder architecture successfully captures the underlying dependency structure and is capable of assigning high predictive importance to the true drivers of the target variable. Conversely,

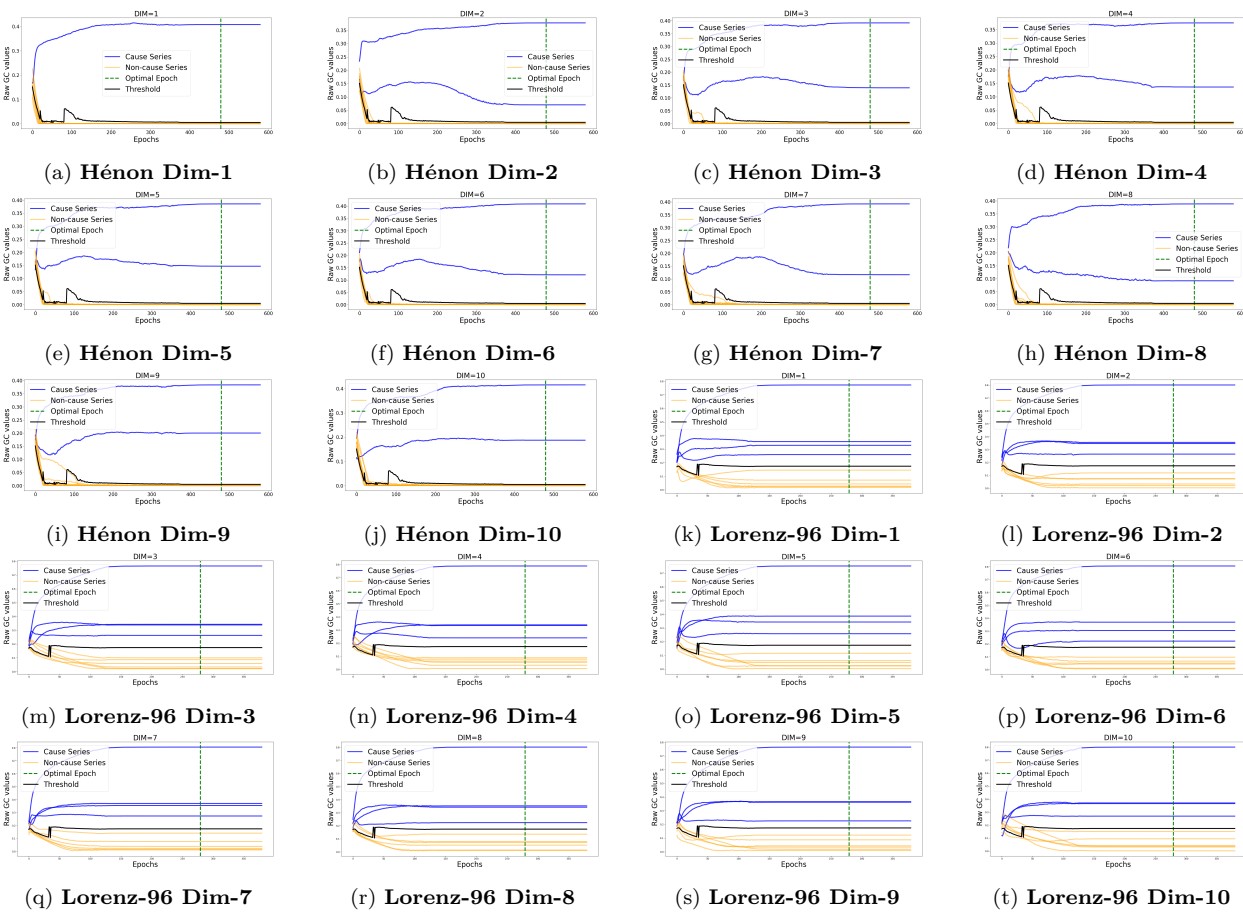

Figure 4: Evolution of learned interaction strengths during training on the Hénon (4a-4j) and Lorenz-96 (4k-4t) dynamical systems. The blue trajectories representing ground-truth causal parents exhibit varying degrees of magnitude but consistently separate from the yellow ones presenting non-causal variables. Black curve represents the evolution of the adaptive threshold at every iteration, which can also be seen getting stabilized. The vertical dashed green line marks the epoch with the minimum validation loss, which is selected as the optimal checkpoint for final graph inference.

the weights corresponding to non-causal inputs are progressively suppressed towards zero. This behavior empirically verifies the effectiveness of the Group Lasso penalty term $\lambda \sum \|W_{(j)}^i\|_2$ in Equation 2 objective function. Therefore, it enforces the necessary sparsity required for Granger causal identification.

**Dynamics of Adaptive Thresholding.** A critical contribution of our framework is the unsupervised selection of the decision boundary $\tau^*$. Instead of a predefined static threshold, $\tau^*$ dynamically adjusts to the changing empirical distribution of the learned weights. As the training progresses and the bimodal distributions diverge, $\tau^*$ settles into a stable region that effectively separates the two modes. These visualizations confirm that our thresholding strategy provides a robust, data-driven alternative to heuristic cutoffs, ensuring that the final adjacency matrix $\hat{A}$ accurately reflects the structural independence assumptions of the ICM principle. On NL-VAR, the trajectories of causal and non-causal weight norms exhibit a more gradual separation than on Hénon and Lorenz-96, reflecting the heterogeneous parent coupling and the wider range of effect sizes induced by the per-edge sampling of $b_{ij}$ and $c_{ij}$. The adaptive threshold tracks this slower divergence and stabilizes once the two modes are established. For CF-Diamond, despite the limited number of weight groups arising from $D = 4$, the convergence trajectories show a clean separation between causal and non-causal weights within the first few hundred epochs, and the threshold settles rapidly into the gap between the two modes.

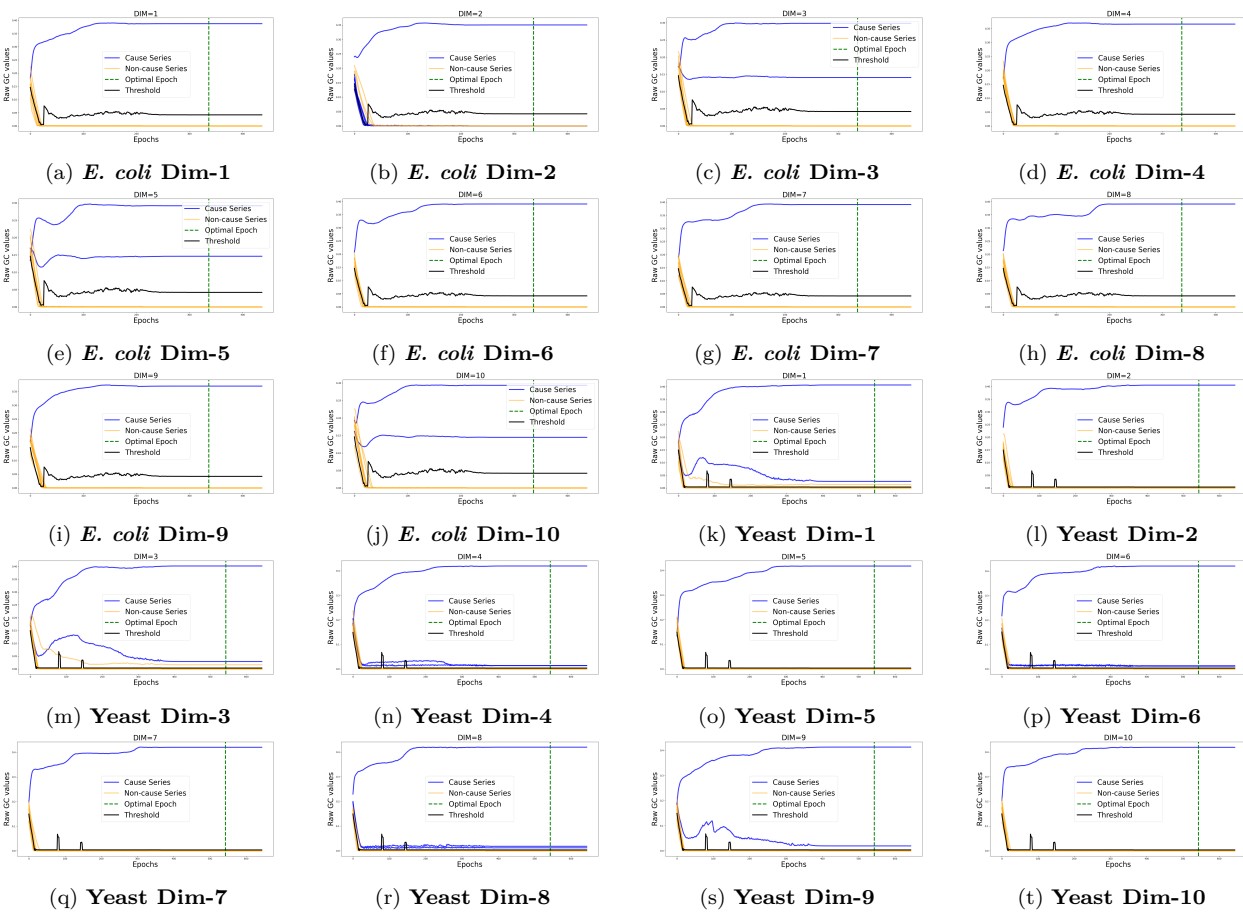

Figure 5: Convergence dynamics of causal discovery on the in silico *E. coli* (5a-5j) and Yeast (5k-5t) gene regulatory benchmarks. Similar to the synthetic experiments, the model successfully disentangles true regulatory interactions (blue) from spurious correlations (yellow). Except for some of the dimensions, the adaptive threshold (black) effectively delineates the causal mechanism despite the higher complexity and noise inherent in biological simulations. The vertical dashed green line marks the epoch with the minimum validation loss, which is selected as the optimal checkpoint for final graph inference.

## F.2 Qualitative Analysis of Causal Graph Estimation

We provide a qualitative visualization of causal graph recovery for all six datasets. For each dataset, we present three matrices: the ground-truth causal adjacency matrix, the estimated raw causal matrix obtained from the learned group-lasso decoder weights prior to thresholding, and the final binary causal graph after applying the proposed adaptive thresholding procedure. The raw causal matrices show that the learned interaction strengths concentrate around the true causal structure. The low-magnitude spurious connections are removed via adaptive thresholding. For the synthetic datasets, the proposed method successfully recovers the underlying causal graphs with minimal structural error. For NL-VAR, the raw matrix exhibits a banded pattern aligned with the random sparse parent structure, and adaptive thresholding recovers the majority of causal links with a small residual structural distance, reflecting the heterogeneity of effect sizes across edges. For CF-Diamond, the raw matrix displays a clean three-band structure corresponding to the diagonal self-loops and the two layers of cross-variable links in the diamond topology; the small graph size amplifies the impact of any single misclassified edge on the F1-score, although the structural recovery remains close to the best-performing baselines. For the in silico biological benchmarks, the recovered graphs exhibit a small number of false positives and false negatives. This behaviour is expected due to measurement noise and the complex, partially overlapping regulatory mechanisms that characterize biological systems. For the Yeast dataset, several causal links are expressed through weak interaction strengths. The adaptive thresholding

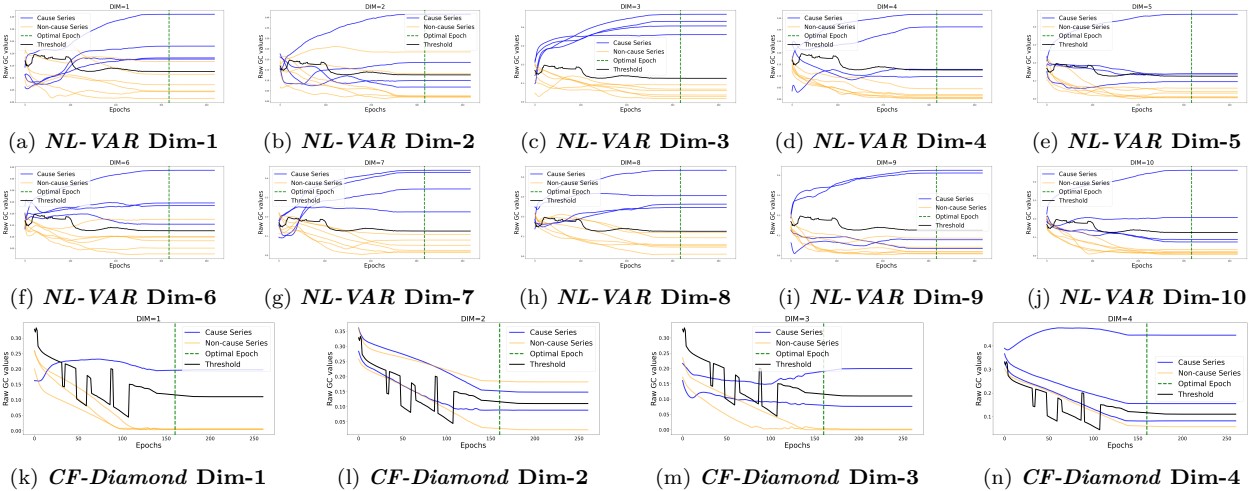

Figure 6: Convergence dynamics of causal discovery on the NL-VAR (6a-6j) and CF-Diamond (6k-6n) benchmarks. The blue trajectories represent ground-truth causal parents, which separate from the yellow non-causal trajectories as training progresses. The black curve denotes the evolution of the adaptive threshold $\tau^*$, which stabilizes once the bimodal distribution of decoder weight norms is established. The vertical dashed green line marks the epoch with the minimum validation loss, selected as the optimal checkpoint for final graph inference.

mechanism successfully recovers these links by separating weak but consistent causal signals from noise. Overall, these visualizations highlight the role of adaptive thresholding and representation-level modularity in producing stable and interpretable causal graphs across diverse dynamical regimes.

### F.3 Analysis of Decoder Weight Distributions

To further validate the adaptive thresholding mechanism proposed in Section 4.3 and to provide qualitative insights into the learned causal representations, we analyze the distribution of decoder weight group norms at the optimal checkpoint across all six datasets.

**Posterior Density Analysis of Decoder Weight Norms.** Figure 8 visualizes the probability density functions of the two-component Gaussian Mixture Model fitted to the empirical distribution of off-diagonal interaction strengths $s^i_{(j)} = \|W^i_{(j)}\|_2$ for each dataset. The upper panel displays the fitted noise and signal component densities, while the lower panel shows a strip plot of individual weight norms colored by their ground-truth causal status.

For the Hénon and Lorenz-96 synthetic benchmarks, the noise and signal densities are well-separated with negligible overlap, and every individual weight norm falls on the correct side of $\tau^*$, consistent with perfect AUROC and SHD scores reported in Table 1. The adaptive threshold adjusts to dataset-specific weight scales. For the Hénon, the threshold value is $\tau^* = 0.0049$ where the group lasso drives non-causal weights (81 out of 100 for a 10-dimensional dataset) to near-zero magnitudes. The threshold value $\tau^* = 0.1743$ for Lorenz-96 shows that the denser causal structure (40 out of 100 for a 10-dimensional dataset) requires larger weight norms to encode the richer set of dependencies.

For the biological benchmarks, the noise and signal densities exhibit greater proximity, and the strip plots reveal that some causal and non-causal norms lie near the decision boundary. This behaviour arises from two properties of these datasets. First, the regulatory dynamics in *E. coli* and Yeast are governed by stochastic kinetic models with thermodynamic noise that attenuates the observable effect sizes of true causal links, resulting in weaker interaction strengths that are harder to distinguish from residual non-causal weights. Second, the transcriptional regulatory networks contain indirect regulatory pathways and partially overlapping regulatory targets, where a single transcription factor simultaneously regulates multiple genes. This induces correlated weight norms across decoder groups sharing the same source variable, broadening the signal component variance and reducing the separation margin from the noise component. Despite this, the

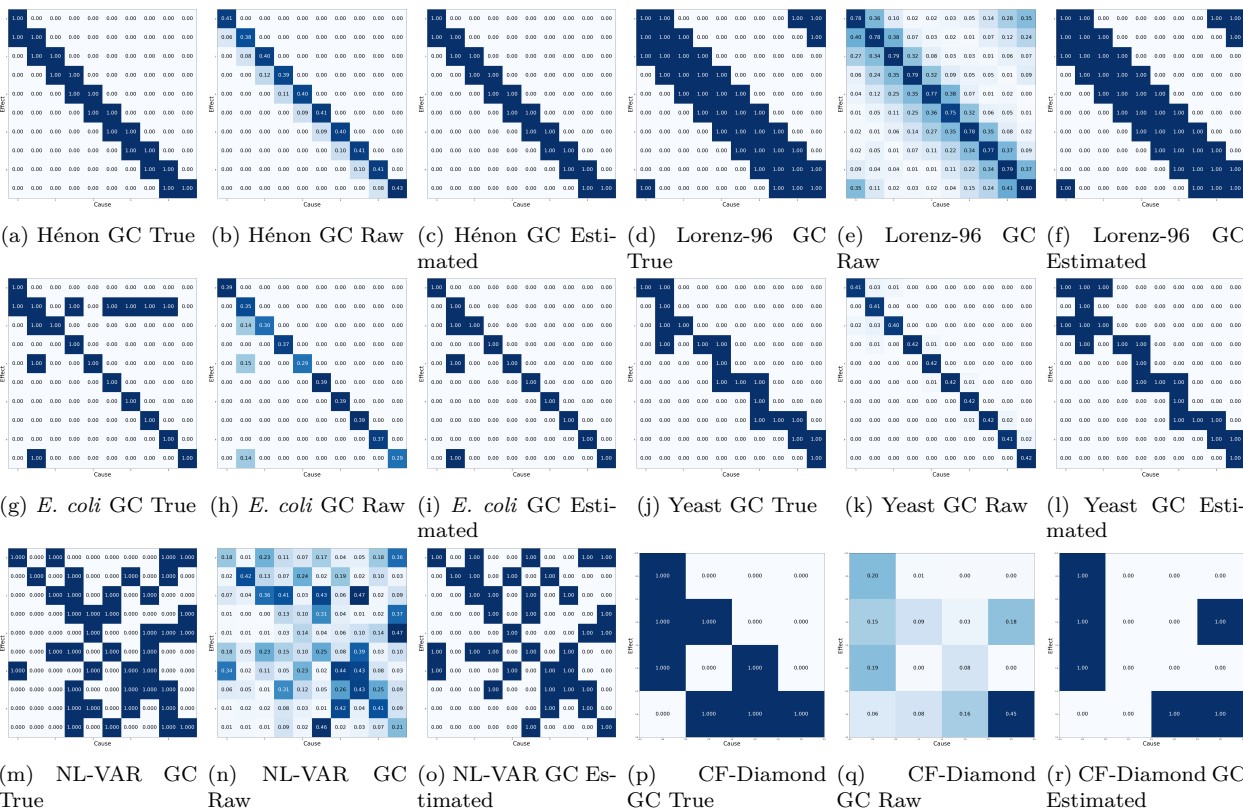

Figure 7: Qualitative visualization of causal graph recovery across all six datasets. For each dataset, we show the ground-truth causal adjacency matrix (left), the estimated raw causal matrix derived from decoder weight norms before thresholding (middle), and the final binary causal graph obtained after applying the proposed adaptive thresholding strategy (right).

GMM correctly identifies the bimodal structure, and the adaptive threshold recovers the majority of causal links. The small number of misclassifications (SHD of 6 for *E. coli* and 7 for Yeast) correspond precisely to weight norms in the overlap region between the two fitted densities. For NL-VAR, the noise and signal densities exhibit moderate overlap, intermediate between the synthetic and biological cases. This is consistent with the heterogeneous parent coupling, where causal effect strengths span a wide range due to the per-edge sampling of $b_{ij}$ and $c_{ij}$, which broadens the signal component. Despite this overlap, the GMM correctly separates the bulk of causal and non-causal weight norms, yielding the F1-score and SHD reported in Table 1. For CF-Diamond, the two components are cleanly bimodal with negligible overlap despite the small number of weight norms available (16 in total), confirming that the factorized encoder produces interaction strength distributions amenable to mixture-based separation even in the small-graph regime.

**t-SNE Visualization of Decoder Weight Groups.** Figure 9 presents t-SNE projections of the full decoder weight group vectors $W^i_{(j)} \in R^H$ at the optimal checkpoint, with each point representing a source-target pair ($j \rightarrow i$) and colored by its ground-truth causal status. Unlike the scalar norm analysis above, this visualization captures the full geometric structure of the learned weight vectors in the latent space.

For Hénon, the non-causal weight groups form a tight central cluster near the origin, while causal weight groups are projected to distant, isolated positions. This clear spatial separation reflects the strong suppression of non-causal weights by the Group Lasso penalty and is consistent with perfect graph recovery. For Lorenz-96, causal weight groups are distributed around the periphery of the embedding space, with non-causal groups concentrated in the centre. Notably, weight groups sharing the same source or target variable tend to cluster in proximity (e.g., $5 \rightarrow 5$, $5 \rightarrow 6$, $4 \rightarrow 5$), suggesting that the factorized encoder produces consistent variable-specific representations that are preserved through the decoder weights.

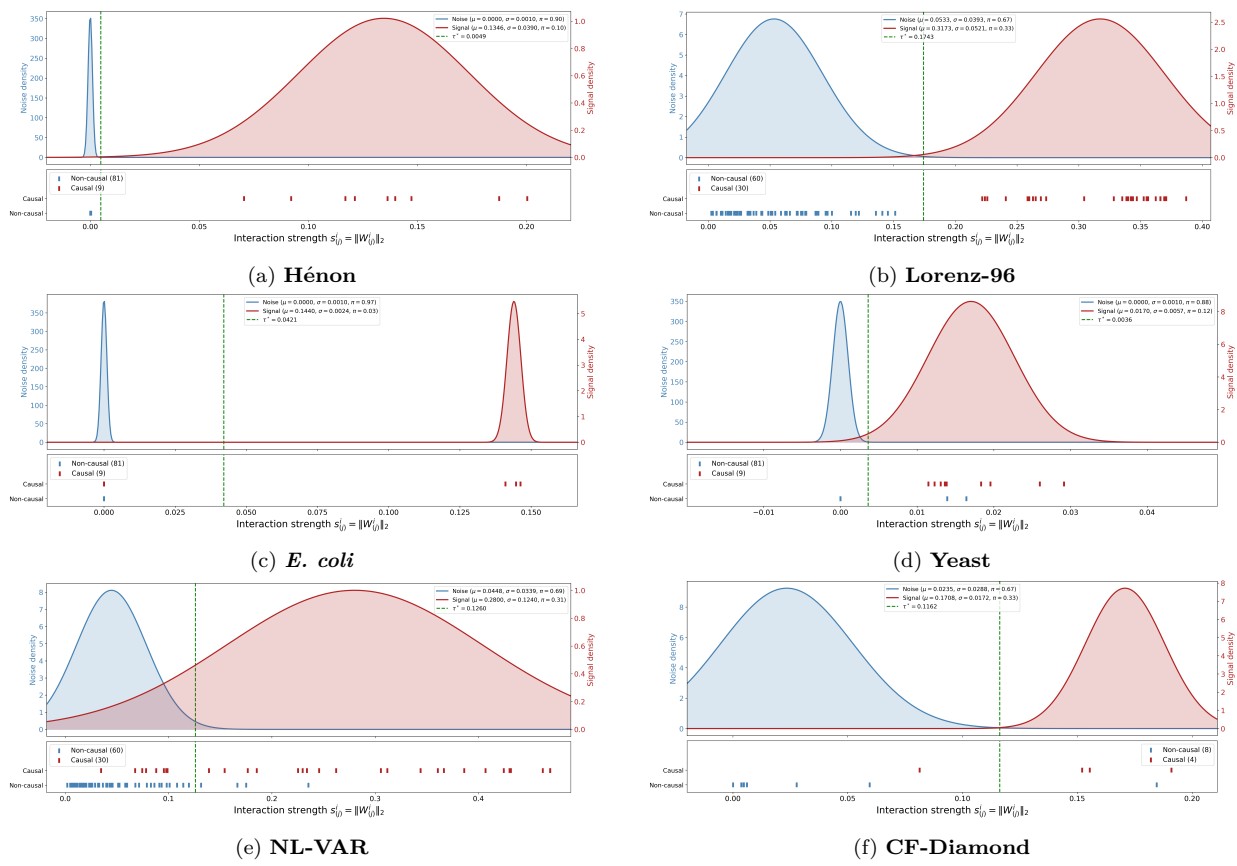

Figure 8: Empirical validation of the bimodal assumption underlying the adaptive thresholding strategy (Equation 3). For each dataset, the upper panel shows the fitted noise and signal probability density functions on dual axes, with the decision boundary $\tau^*$ computed via posterior intersection (Equation 4). The lower panel displays individual off-diagonal weight norms, stratified by ground-truth causal status.

For the biological benchmarks, the separation between causal and non-causal weight groups is less pronounced but still discernible. In *E. coli*, causal weight groups are generally positioned at the boundary of the non-causal cluster, with edges from the same regulatory hub (e.g., edges originating from variable 1) forming localized sub-clusters. In Yeast, the causal and non-causal groups exhibit greater spatial intermixing, consistent with the narrower density separation observed in the posterior density analysis and the higher SHD reported in Table 1. On NL-VAR, the t-SNE projection exhibits a more dispersed structure than on the synthetic systems, with causal weight groups distributed across multiple regions of the embedding space rather than clustered at the periphery. This is consistent with the heterogeneous coupling coefficients $b_{ij}$ and $c_{ij}$ sampled per parent edge, which produce variable-specific weight magnitudes that reflect the strength of each individual causal effect. For CF-Diamond, despite the small number of weight groups (16 in total), causal and non-causal points occupy clearly distinct regions of the embedding, confirming that the factorized encoder produces well-separated weight representations even when the empirical distribution available for visualization is sparse. These observations corroborate the quantitative findings: the difficulty of causal discovery in biological systems arises not from architectural limitations but from the inherently weaker and more heterogeneous causal signals in these datasets.

## G  Implementation setup

The whole system has been implemented in PyTorch v2.4.0 with support for CUDA v12.5 and trained on a single NVIDIA RTX 5000 Pro GPU with 35GB VRAM. We have chosen Transformer (Vaswani et al., 2017) for implementing the encoder heads of the main causal recurrent model. We have set the values of $\beta$ and

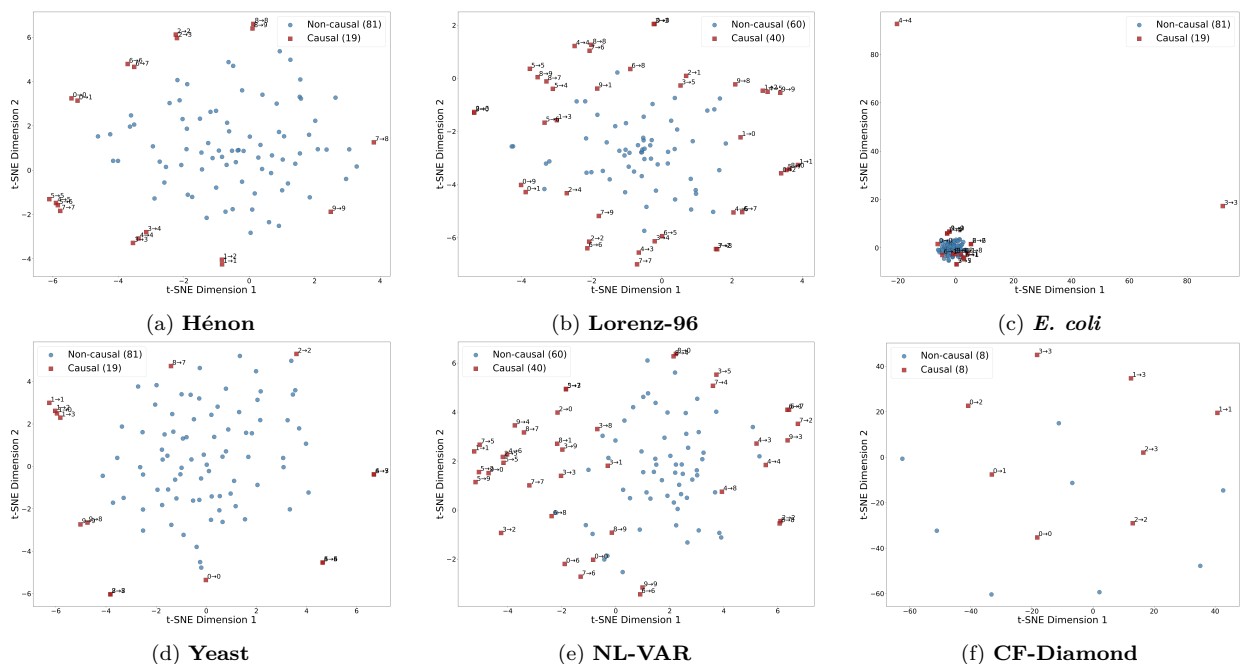

(a) **Hénon**  (b) **Lorenz-96**  (c) *E. coli*

(d) **Yeast**  (e) **NL-VAR**  (f) **CF-Diamond**

Figure 9: Geometric structure of learned decoder weight group vectors $W_{(j)}^i \in \mathbb{R}^H$ at the optimal checkpoint visualized via t-SNE. Each point represents a source-target pair, with causal edges (annotated red squares) and non-causal edges (blue circles). Causal weight groups occupy distinct regions of the embedding space, with separation quality correlating with graph recovery performance.

$\lambda$ as 0.05 and 0.1, respectively. For AdamW optimizer, we keep the default values of hyperparameters as implemented in PyTorch v2.4.0. The maximum number of epochs is 2000, and the learning rate $\gamma$ has been set at 0.001. Additionally, we have incorporated early stopping regularization based on the validation loss, which stops the training if the validation loss does not decrease for 200 consecutive epochs. Table 4 reports

Table 4: Quantitative comparison of runtime and memory consumption on the Hénon dataset.

| Methods | Total training time (m) | Time per Epoch (s) | Peak GPU VRAM Consumption (MB) |
|---|---|---|---|
| CausalFormer (Kong et al., 2025) | 56.2 ($\pm$ 3.8) | 7.80 ($\pm$ 0.40) | 3654 ($\pm$ 25) |
| CR-VAE (Li et al., 2023) | 30.25 ($\pm$ 2.5) | 5.66 ($\pm$ 2.5) | 2052 ($\pm$ 10) |
| NGC (Tank et al., 2021) | 28.44 ($\pm$ 1.9) | 4.26 ($\pm$ 1.7) | 1947 ($\pm$ 9) |
| GVAR (Marcinkevičs & Vogt, 2021) | 12.42 ($\pm$ 1.22) | 1.55 ($\pm$ 0.2) | 566 ($\pm$ 4) |
| TCDF (Nauta et al., 2019) | 18.69 ($\pm$ 1.65) | 2.31 ($\pm$ 0.48) | 1026 ($\pm$ 7) |
| **CauFR-TS (Proposed)** | 20.14 ($\pm$ 1.15) | 2.65 ($\pm$ 0.55) | 1504 ($\pm$ 6) |

the total training time, time per epoch, and peak GPU VRAM consumption of all methods on the Hénon dataset, measured over 5 runs. Computational costs are representative across all benchmarks, as all datasets share the same dimensionality ($D = 10$) and sequence length ($T = 50$). CauFR-TS incurs a total training time of 20.14($\pm$1.15) minutes and a peak memory consumption of 1504($\pm$6) MB, which is moderate relative to the full baseline spectrum. The higher cost of CausalFormer (56.2 minutes, 3654 MB) is attributable to its global self-attention mechanism operating over the full multivariate input, whose complexity scales quadratically with both sequence length and the number of variables. CR-VAE and NGC are costlier than CauFR-TS despite relying on simpler overall designs, because both employ a shared encoder that processes the full $D$-dimensional input at each step alongside multi-head RNN-based decoder heads dedicated to each of the $D$ target variables, resulting in denser parameter interactions and larger activation maps throughout the forward pass. In contrast, CauFR-TS replaces the recurrent decoder heads with lightweight single-layer MLP heads, which significantly reduces the per-step computational cost of the decoding stage and offsets

the overhead introduced by the $D$ independent encoder heads. GVAR and TCDF are lighter due to their comparatively shallower architectures and the absence of variational components. While CauFR-TS employs $D$ independent encoder heads, each head processes only a univariate series of dimensions $\tau \times 1$ rather than $\tau \times D$, making each individual encoder significantly lighter than an equivalent shared encoder. The total computational footprint is therefore comparable to a single shared encoder of the same aggregate capacity, and the factorized design does not introduce disproportionate overhead relative to the performance gains achieved.

