# OpenReview forum: "CauFR-TS: Causal Time-Series Identifiability via Factorized Representations"
_TMLR — Accepted by TMLR_

### Review · Reviewer_zriU · 2026-03-11

**Summary Of Contributions:**

This work contributes to the field of time-series Granger-causal discovery. The main contributions the authors make are twofold: 1) They characterize a pitfall of approaches that encode input time series to a shared latent space, which may lead to inducing unobserved confounding and suggest a factorized encoder architecture that leads to latent variables that only depend on single observed variables, 2) They propose an adaptive approach to set the threshold above which weights in the decoder are considered significant enough to be interpreted as existing causal connections between variables. The authors propose to fit a bimodal Gaussian Mixture Model over the learned decoder weights, elliminating the need for a hand-tuned threshold hyperparameter.

Additionally, experimental results on synthetic and semi-synthetic data is presented, where the merits of the proposed method against competing approaches is demonstrated empirically.

**Audience:**

Yes

**Audience Explanation:**

This work is concerned with time series Granger-causal discovery, for which there exists a community with a number of recent works. The approach proposed by the authors extends these works in novel ways, meriting interest from this community.

**Claims And Evidence:**

Yes

**Claims Explanation:**

Overall, I believe the claims made in this paper are supported by evidence. The assumptions made are sensible in the context of time series causal discovery (factorized representation) and the proposed method seems to outperform competing approaches. The main ideas are nicely presented and the paper is written and organized in way that makes following easy.

I do however believe that the claims that are made could be more clear, as I find that main claim that learning representations *necessarily* leads to unobserved confounding not totally convincing. I expand upon this in the Requested Changes section.

**Requested Changes:**

1. Discussion of the main claim of necessity of representation learning leading to unobserved confounding. I believe this point to mainly be an issue of presentation and framing. Most works on (causal) representation learning assume something about the latent variables, which is the target of learning: ICA aims to learn independent latents from mixed observations, causal representation learning causally related latents from entangled measurements. Given this context, it is surprising if the observations you assume to have access to already fulfil the desired property: they are the causal variables over which we aim to learn a graph. If one were to be in the "typical" representation learning case I describe, I don't at all see why passing observations through an encoder necessarily leads to dependencies between variables. If this were true, how would ICA ever be able to succeed? Additionally, in your setting, wouldn't an encoder that learns the identity map not suffer from the issues put forward in your proposition? Or is this case ruled out by your assumptions?
2. Could you provide error bars or confidence intervals for the experimental results?

### Minor Points
- There is lots of work on time series causal discovery not based on Granger causality, which would be worth mentioning and citing. See [1] for pointers on relevant work.
- Some technicalities like the temporal graph or the causal matrix $A$ could be introduced and defined more rigorously.
- The Varici et al., 2024 reference seems to have a typo.
- Multiple references are missing a venue. Are these perhaps arXiv papers?
- Broken reference in the last paragraph of Section 6.2

The point 1. is important to me and should be discussed in order for me to recommend acceptance. The others would strengthen the work.

#### References
[1] Hasan, U., Hossain, E., & Gani, M. O. A Survey on Causal Discovery Methods for IID and Time Series Data. Transactions on Machine Learning Research.

---

> ### Author Response · Authors · 2026-03-27
>
> We sincerely thank the reviewer for the thoughtful and constructive feedback. The reviewer’s positive assessment of the paper’s organization, the soundness of the factorized representation idea, and the merits of the adaptive thresholding contribution are greatly appreciated. We are particularly grateful for the sharp theoretical question raised in Point 1, which touches on a genuinely important subtlety in the framing of our claims. We believe that addressing it carefully has meaningfully strengthened the paper. Below, we respond to each point in turn and outline the concrete changes we made in the revision.
> ___
> **Response to Point-1 -- On the unobserved latent confounding in shared encoders:**
> We thank the reviewer for this precise and insightful challenge. We agree that the framing of our claim deserves clarification, and we have **revised Section 3** accordingly in the updated manuscript. Let us address the sub-questions raised.
>
> **1. On the ICA analogy:** ICA succeeds because it imposes statistical independence of latents as an explicit optimization objective where independence is the structural constraint driving the solution. In our setting, the shared encoder is optimized for predictive accuracy, not independence. The optimizer therefore exploits all available correlations in the data to minimize forecasting error, and there is no force in the objective that would drive the encoder toward a block-diagonal Jacobian structure. Our Propositions 1 and 2 apply precisely in this regime.
>
> **2. On the identity map and scope of our claims:** The identity map is a degenerate edge case that is never architecturally guaranteed to emerge from gradient-based training on correlated multivariate data. More importantly, our claim is not that every shared encoder empirically fails in every instance; rather, it is that there exists no architectural guarantee preventing entanglement. Causal identifiability requires structural guarantees, not empirical tendencies that are contingent on data geometry and optimization dynamics. CauFR-TS provides this guarantee by construction, as formalized in **Remark 1 (Section 6.1, Page 9)**, followed by a detailed discussion in **Appendix D.3**.
>
> **3. Proposed Revision:** We have added a clarifying remark after **Proposition 2 in Section 3**, making this distinction explicit, along with a **technical note in Appendix D.2** formalizing the identity map edge case.
> ___
> **Response to Point 2 -- Error Bars and Confidence Intervals:** We agree that reporting statistics across multiple runs is essential for rigorous empirical assessment. We have re-run all experiments with 5 random seeds and reported the mean $\pm$ standard deviation in the revised manuscript **(Table 1, Page 10)**. The relative performance of CauFR-TS over all baselines remains consistent across seeds.
> ___

---

> > ### Author Response · Authors · 2026-03-27
> >
> > (continuation)
> >
> > **Response to Minor Points:**
> > - **Non-Granger causal discovery literature.** We thank the reviewer for this suggestion. We have added a brief discussion in the first paragraph of Introduction acknowledging non-Granger approaches to time-series causal discovery, including constraint-based [1,2,3] and score-based methods [4,5], and cited the recommended survey [6] alongside other relevant works.
> > - **Rigorous definition of temporal graph and causal matrix A.** We agree these should be defined more formally. We have added an explicit brief notational clarification of the temporal causal graph $G$ and the adjacency matrix $A$ after **Definition 1 in Section 2 (Page 3)**.
> > - **Typo in the citation.** Thank you for catching this. We have corrected it in the revision.
> > - **Missing venues.** We have verified that all references with missing venue information are arXiv preprints. The TMLR submission formatting does not render the arXiv repository as a venue field in the reference list. We have explicitly annotated those as preprints in the revised manuscript to avoid ambiguity.
> > - **Broken reference.** This unresolved LaTeX reference has been fixed to point to the correct appendix section **(Appendix F)**. We have verified that no other broken references remain in the revised manuscript.
> > ___
> > We thank the reviewer once more for the careful and constructive engagement with our work. The proposed revisions and minor comments have collectively strengthened the paper. We hope these changes adequately address the reviewer’s concerns and look forward to further feedback.
> > ___
> > **References**
> >
> > [1] Shanyun Gao, Raghavendra Addanki, Tong Yu, Ryan A. Rossi, and Murat Kocaoglu. Causal discovery in semi-stationary time series. In Proceedings of the 37th International Conference on Neural Information Processing Systems, NIPS ’23, Red Hook, NY, USA, 2023. Curran Associates Inc.
> >
> > [2] Jakob Runge. Discovering contemporaneous and lagged causal relations in autocorrelated nonlinear time series datasets. In Jonas Peters and David Sontag (eds.), Proceedings of the 36th Conference on Uncertainty in Artificial Intelligence (UAI), volume 124 of Proceedings of Machine Learning Research, pp. 1388–1397. PMLR, 03–06 Aug 2020.
> >
> > [3] Jakob Runge, Peer Nowack, Marlene Kretschmer, Seth Flaxman, and Dino Sejdinovic. Detecting and quantifying causal associations in large nonlinear time series datasets. Science advances, 5(11):eaau4996, 2019.
> >
> > [4] Xiangyu Sun, Oliver Schulte, Guiliang Liu, and Pascal Poupart. Nts-notears: Learning nonparametric dbns with prior knowledge. In Francisco Ruiz, Jennifer Dy, and Jan-Willem van de Meent (eds.), Proceedings of The 26th International Conference on Artificial Intelligence and Statistics, volume 206 of Proceedings of Machine Learning Research, pp. 1942–1964. PMLR, 25–27 Apr 2023.
> >
> > [5] Roxana Pamfil, Nisara Sriwattanaworachai, Shaan Desai, Philip Pilgerstorfer, Konstantinos Georgatzis, Paul Beaumont, and Bryon Aragam. Dynotears: Structure learning from time-series data. In Silvia Chiappa and Roberto Calandra (eds.), Proceedings of the Twenty Third International Conference on Artificial Intelligence and Statistics, volume 108 of Proceedings of Machine Learning Research, pp. 1595–1605. PMLR, 26–28 Aug 2020.
> >
> > [6] Uzma Hasan, Emam Hossain, and Md Osman Gani. A survey on causal discovery methods for i.i.d. and time series data. Transactions on Machine Learning Research, 2023. ISSN 2835-8856. Survey Certification.

---

### Review · Reviewer_78uA · 2026-03-13

**Summary Of Contributions:**

Neural causal discovery methods for multivariate time series often use shared latent encoders, which entangle different causal mechanisms and lead to non-identifiability and violations of conditional independence assumptions needed for reliable Granger-causal structure learning. The paper proposes CauFR-TS, a recurrent variational framework with dimension-wise encoders and structured latent aggregation to enforce modular causal mechanisms and properly mediate cross-variable dependencies. It introduces an adaptive, unsupervised link selection method based on decoder weight distributions (instead of heuristic thresholds), achieving better causal graph recovery on synthetic and biological benchmarks while maintaining competitive forecasting performance.

**Audience:**

Yes

**Audience Explanation:**

Granger causality, time series forecasting, latent factorization, and identifiability are all important research fields. This paper tries to resolve the theoretical limitations of the shared encoders that entangle distinct causal mechanisms into a unified latent manifold.

**Broader Impact Concerns:**

/

**Claims And Evidence:**

No

**Claims Explanation:**

See the "request changes" section.

**Requested Changes:**

* In Def 1, $x^{(i)} \to x^{(i)}$ seems to be a typo.
* In Def 4, what is $p$ and what is $D$. I can guess them, but they should be formally defined or explained.
* Before Sec 4.2, "Taken together, the decoder parameterizes the joint conditional distribution $p(x_T |x_{1:T −1})$". I believe this is not only regarding the decoder but also the encoder.
* I find the derivation and presentation before the experiment section hard to follow. It's not clear to me which is the core contribution and which is the technical design from the authors. For example, Eq. 2 seems to be a common way VAE loss function with sparsity. I think authors should make the logic better and focus more on the technical contributions, math derivations, rather than plainly list the definitions, formulae, and algorithms. Also, the notation should be clean, rigorous, and non-verbose.
* For the result part, I understand that these commonly used metrics are good. But latent visualization and interpretation are also important as a qualitative way of evaluating different methods. I strongly recommend that the authors do more exploratory analysis on the results and explain what the learned parameter means for each given dataset.
* Before Sec 6.3, there is a sentence like "value in Appendix ??". Given this is a journal submission which has no submission deadlines, these typos without proofreading is a little bit unacceptable. It is not plain to run some experiments and put the metrics, and then say your method is good. It is important to illustrate and introduce the whole story in a valuable way.

---

> ### Author Response · Authors · 2026-03-27
>
> We sincerely thank the reviewer for the careful reading of our manuscript and the constructive feedback. The comments on presentation clarity, notational rigour, and the need for deeper exploratory analysis have been particularly valuable in improving the paper. We have thoroughly revised the manuscript to address each concern, including correcting all typographical and referencing errors, clarifying the distinction between our core contributions and standard components, and adding new qualitative analyses of the learned representations. Below, we respond to each point in detail.
> ___
> **1. Notation and Proofreading Corrections:** We appreciate the reviewer for identifying these issues and sincerely apologize for the oversights, which are unacceptable in a journal submission. We have conducted a thorough proofreading pass of the entire manuscript. The specific corrections are as follows:
> - **Definition 1:** The causal link direction $x^{(i)} \rightarrow x^{(i)}$ was indeed a typo. This has been corrected to $x^{(j)} \rightarrow x^{(i)}$, consistent with the definition’s statement that $x^{(j)}$ Granger causes $x^{(i)}$.
> - **Definition 4:** We have added explicit scoping for both $p$ and $D$ within the definition itself. Specifically, $D$ denotes the dimensionality of the multivariate time series $X_t = \\{x_t^{(1)}, \ldots, x_t^{(D)} \\}$ and $p \in \\{1, \ldots, D\\}$ indexes the individual component variables. The revised definition is now self-contained in the updated manuscript.
> - **The conditional distribution statement in Section 4.1:** The reviewer is correct that the conditional distribution $p(x_T|x_{1:T-1})$ is parameterized jointly by both the encoder and decoder. The encoder models the variational posterior $q_\Phi (z|x_{1:T-1})$ by mapping each variable’s history to its latent representation, while the decoder models the likelihood $p_\Theta (x_T|z)$ by mapping the aggregated latent representation to the predictive distribution. We have revised the sentence to the following: *“Taken together, the encoder-decoder pipeline jointly parameterizes the conditional distribution $p(x_T|x_{1:T-1})$."*
> - **Broken Reference:** This unresolved LaTeX reference has been fixed to point to the correct appendix section **(Appendix F)**. We have verified that no other broken references remain in the manuscript.
> ___
> **2. Presentation Clarity and Distinguishing Core Contributions:** We thank the reviewer for this important observation. We agree that the manuscript should more clearly delineate which components are novel contributions and which are standard building blocks adopted for our framework. We have revised the presentation accordingly.
>
> We wish to clarify that the novelty of CauFR-TS does not lie in the objective function itself. As the reviewer correctly notes, Equation 2 is a standard VAE loss augmented with group-lasso sparsity, which is intentional. The central contribution of our work is the formal demonstration that applying this same, well-understood objective to architectures with shared encoders yields structurally non-identifiable causal graphs **(Section 3, Propositions 1–2)** and that replacing the shared encoder with a factorized architecture restores identifiability under the same objective **(Remark 1, Page 9)**. Concretely, our core contributions are as follows:
> - **Theoretical analysis (Section 3):** Our contribution is not the loss function but the formal demonstration that this same objective yields non-identifiable causal graphs under shared encoders **(Propositions 1–2, with detailed proofs provided in Appendix D)** and that factorized encoding restores identifiability **(Remark 1, elaborated in Appendix D.3)**.
> - **Factorized encoder architecture (Section 4.1):** The channel-wise encoder design ensures a block-diagonal Jacobian structure, making the decoder weight groups $W^i_{(j)}$ the exclusive gateway for inter-variable information flow. This is what makes the standard group-lasso penalty identifiable.
> - **Adaptive GMM-based thresholding (Section 4.3):** A parameter-free causal link selection strategy that replaces heuristic cutoffs with a data-driven decision boundary.
> ___
> (continued)

---

> > ### Author Response · Authors · 2026-03-27
> >
> > (continuation)
> > ___
> > **3. Latent Visualization and Exploratory Analysis:** We appreciate this suggestion. The original submission already includes substantial exploratory analysis in Appendix F: dimension-wise convergence plots showing the stabilization of group-lasso decoder weight norms alongside the evolving adaptive threshold across training epochs **(Figures 4–5, Pages 24-25)** and qualitative visualizations of the ground-truth, raw continuous, and thresholded binary causal adjacency matrices for all four datasets **(Figure 6, Page 26)**. Together, these illustrate how the learned interaction strengths separate causal from non-causal links and how the adaptive threshold converts the continuous causal matrix into the final binary graph.
> >
> > In the revised manuscript, we have additionally included:
> > - Visualizations of the posterior density of decoder weight norms for each dataset **(Figure 7, Page 26)**, showing the fitted noise and signal components overlaid on the empirical distribution of decoder weight norms, along with the computed threshold $\tau^{\*}$ confirming that the bimodal assumption holds empirically and illustrating how $\tau^{\*}$ adapts to dataset-specific weight scales.
> > - t-SNE visualizations of the decoder weight groups **(Figure 8, Page 27)** at the optimal checkpoint for all four datasets, showing clear clustering separation between causal and non-causal weights.
> >
> > These analyses have been added to **Appendix F, with a summary discussion in Sections 4.3 and 6.1**, directing readers to these results.
> > ___
> > We believe the above revisions comprehensively address the reviewer’s concerns and have meaningfully strengthened the manuscript. We are grateful for the feedback and remain open to further suggestions.

---

### Review · Reviewer_eXtQ · 2026-03-16

**Summary Of Contributions:**

Motivated by the proposition that a shared latent representation can produce entangled representations that prevent variable-wise separation and violate the Independent Causal Mechanisms principle, the paper proposes a recurrent variational framework that enforces mechanism independence through channel-wise encoders and models cross-variable dependencies via structured latent aggregation. It also introduces an adaptive, data-driven thresholding procedure to determine edges from the continuous weights in the estimated causal adjacency matrix.

The paper provides propositions showing that shared latent representations can induce confounding among source variables rather than preserve the factorization implied by Independent Causal Mechanisms. It further shows that the proposed algorithm avoids this limitation through the use of channel-wise encoders. Empirically, the paper demonstrates strong performance on four datasets against multiple baselines.

**Audience:**

Yes

**Audience Explanation:**

The paper focuses on an interesting and important question: whether a shared encoder introduces confounding and consequently leads to an incorrect (denser) causal graph. Across four commonly used datasets, the proposed algorithm achieves impressive performance through the use of a channel-wise decoder and an adaptive thresholding procedure for edge selection.

The overall framework of the paper is complete and includes an ablation study.

**Broader Impact Concerns:**

No concerns.

**Claims And Evidence:**

No

**Claims Explanation:**

There are three main concerns:

1. The motivation could be better justified. While the Independent Causal Mechanisms principle must be imposed on the raw data, it is not clear why it should also be enforced in the latent space after a nonlinear transformation. The paper would benefit from a clearer explanation of the rationale behind the channel-wise encoder and its potential advantages.

2. The two propositions are not supported by fully rigorous proofs, which makes the above concern more significant. The claim that a shared encoder introduces confounding and consequently leads to an incorrect causal graph should be established more rigorously.

3. A more extensive evaluation would be desirable beyond the four datasets currently considered. In particular, simulations on synthetic datasets, with average performance and standard deviations reported, would provide a more comprehensive assessment of the method.

**Requested Changes:**

Please also refer to the previous comment.

A comparison of computational cost and running time would also be helpful.

---

> ### Author Response · Authors · 2026-03-27
>
> We thank the reviewer for the careful reading of our work and the constructive feedback. The reviewer correctly identifies the core contributions of the paper and raises three substantive concerns regarding the motivation, theoretical rigour, and empirical evaluation. We address each point below.
> ___
> **1. Motivation for Latent-Space ICM Enforcement:** We thank the reviewer for raising this point. We would like to clarify that our argument is not that ICM must hold in the latent space by definition. Rather, the latent representation is the computational pathway through which Granger causality is estimated. If this pathway entangles signals from multiple variables **(formally shown via Proposition 1 and 2, with proofs in Appendix D)**, the downstream causal estimator cannot recover the conditional independence structure of the true data-generating process, even if that process genuinely satisfies ICM and even if decoder-level sparsity is enforced. The factorized encoder is therefore not an assumption about the data, but a structural design choice that preserves the ICM factorization through the estimation pipeline, ensuring the causal estimator operates on deconfounded representations.
>
> In the revised manuscript, we have added a **clarifying paragraph at the beginning of Section 3** making this distinction explicit.
> ___
> **2. The rigour of Propositions 1 and 2:** We acknowledge that the proofs in the main text are presented as proof sketches for conciseness. However, **Appendix D** provides the formal gradient-level argument underpinning both propositions. Proposition 1 is grounded in the dense Jacobian structure of shared encoders, showing that without explicit architectural constraints, every latent dimension becomes a function of all input channels. Proposition 2 follows by decomposing the total causal gradient into encoder and decoder terms (Equation 8), demonstrating that the non-zero encoder Jacobian creates an irreducible confounding path even when the true causal link is absent.
>
> In the revised manuscript, we have added **an explicit assumptions paragraph** at the outset listing the conditions under which the propositions hold **(Appendix D, Page 19)**.
> ___
> **3. Evaluation Breadth and Reporting of Statistics:** We agree this is a valid and important point. The four datasets were deliberately chosen to represent a spectrum from clean synthetic dynamical systems to noisy biological simulations, providing a meaningful diversity of causal structures and noise regimes. Nevertheless, we agree that reporting statistics across multiple runs is essential for a rigorous empirical assessment. We have re-run all experiments with 5 random seeds and reported the mean $\pm$ standard deviation in the revised manuscript **(Table 1, Page 10)**. The relative performance of CauFR-TS over all baselines remains consistent across seeds.
> ___
> **4. Computational Cost and Running Time:** We appreciate this suggestion. CauFR-TS employs $D$ independent encoder heads, which introduces overhead linear to the number of variables $D$ compared to a single shared encoder. However, each encoder head processes only a univariate time series, making each individual encoder significantly lighter than a shared encoder processing the full multivariate input. In practice, this trade-off is favourable. We report the total training time, average training time per epoch, and peak GPU VRAM usage (all with mean $\pm$ standard deviation) in the revised manuscript **(Appendix G, Table 4, Page 28)**.
> ___
> We believe the above revisions collectively address the reviewer’s concerns. We are confident that these changes further strengthen the paper and hope the reviewer will consider them in their final assessment.

---

### Author Response · Authors · 2026-03-27
**Thanking the Reviewers and the Action Editor**

We thank the Reviewers and the Action Editor for their time and constructive feedback. We have carefully addressed all the comments and clarified the questions raised. All changes are highlighted in blue in the revised manuscript for ease of review.

---

### Decision · Action_Editor_qMnx · 2026-04-21

**Recommendation:** Accept with minor revision

**Audience:**

Yes

**Audience Explanation:**

Causal discovery from multivariate time series is a practical topic in the machine learning area.

**Claims And Evidence:**

Yes

**Claims Explanation:**

The review team agrees that the target research question is important and interesting, and the claims made in this submission are supported. However, more supporting evidence should be added. In the next version, I would like to see the following revisions

1. The two propositions are not established with fully rigorous proofs, even in Appendix D.
2. The evaluation based on four datasets is not comprehensive.

---

> ### Author Response · Authors · 2026-05-01
>
> We thank the Action Editor for the careful assessment and constructive guidance. We have thoroughly addressed both revision requests.
> ___
> **Response to Point 1.** The previous Appendix D contained two informal steps: Proposition 1 relied on an optimizer-behaviour argument, and Proposition 2 used an approximate single-neurone model $(z_{t,h} \approx \sigma(\cdot))$. Both are replaced with formal derivations.
>
> We introduce **Lemma 1 (Generic Density of the Shared Encoder Jacobian, Appendix D, Page 19)**, which establishes that every entry of the encoder Jacobian is non-zero for Lebesgue-almost every parameter-input pair. The proof rests on (i) generic invertibility of activation derivative matrices under Assumption (A1), and (ii) a polynomial non-vanishing argument via [1], showing that each Jacobian entry, being a non-identically-zero polynomial in the parameters, vanishes only on a measure-zero set. This lemma serves as the common formal foundation for both propositions.
>
> The proof of **Proposition 1 (Appendix D.1, Page 20)** now invokes Lemma 1 directly: the generically non-zero off-diagonal Jacobian blocks contradict the block-diagonal structure required for variable-wise separability. The proof of **Proposition 2 (Appendix D.2, Pages 20–21)** is restructured into four steps using an exact inner-product formulation $\frac{\partial \hat{x}^{(i)}\_{t}}{\partial \hat{x}^{(j)}\_{t-}} = \mathbf{d}_i^\top \mathbf{e}_j$, where $\mathbf{d}\_i$ and $\mathbf{e}\_j$ are the decoder and encoder sensitivity vectors. The proof shows that this inner product is a non-identically-zero polynomial via a concrete witness configuration and concludes via the same measure-theoretic argument as Lemma 1. The remark on the identity map (Section D.2, Page 21) and the justification of Remark 1 (Section D.3, Pages 21–22) are retained with **updated back-references to Lemma 1**.
> ___
> **Response to Point 2.** The original four-dataset evaluation did not adequately probe the diversity of graph topologies and dynamical regimes. We have added two synthetic benchmarks, broadening the evaluation along complementary axes of Graph topology and Nonlinearity structure, and evaluated under the same protocol with mean $\pm$ standard deviation across 5 random seeds.
>
> **- CF-Diamond (Section 5.1, Page 8; Appendix A, Pages 16–17).** A four-variable diamond benchmark adopted from [2], combining mediators, a collider, and self-causal links with mixed lagged and instantaneous dependencies. It complements the chain topology of Hénon and the dense topology of Lorenz-96 by probing recovery under overlapping ancestral paths, a canonical failure mode for shared-encoder architectures. Adopting it from a strong baseline (CausalFormer) ensures the comparison favours that baseline.
>
> **- NL-VAR (Section 5.1, Page 8; Appendix A, Page 17).** A 10-dimensional first-order autoregressive process with quadratic parent interactions and a smooth saturating transition on a random sparse topology with heterogeneous per-edge coupling, probing recovery under stochastic saturating nonlinear regimes with non-uniform effect sizes.
>
> **Coverage of the expanded benchmark suite.** The six benchmarks now span chaotic, dissipative, and saturating nonlinear dynamics; deterministic and stochastic processes; chain, dense, diamond, and random topologies; and synthetic and biologically grounded networks. Results appear in Tables 1 and 2 (Page 11); convergence, qualitative recovery, posterior density, and t-SNE analyses on the new benchmarks are in Figures 6–9 (Pages 27–30); expanded discussion is in Section 6.1 (Page 10).
>
> **Empirical findings.** On NL-VAR, CauFR-TS attains the highest AUROC (0.944), highest F1 (0.853), and lowest SHD (12). On CF-Diamond, it achieves perfect ranking (AUROC = 1.0) but a marginally lower F1 (0.659) than CausalFormer and TCDF (0.681); the gap is attributable to the small-graph regime, where only 16 weight norms are available to the GMM, while the perfect AUROC confirms correct edge ranking. CauFR-TS also attains the lowest one-step-ahead RMSE on every benchmark, including the two new systems (Table 2, Page 11).
> ___
> We thank the Action Editor once again for the careful and constructive guidance in the decision. The two requested revisions have substantially improved the rigour of the theoretical analysis and the breadth of the empirical evaluation. We hope the revised manuscript adequately addresses both points and remain happy to incorporate any further feedback.
> ___
> **References**
>
> [1] J. T. Schwartz. 1980. Fast Probabilistic Algorithms for Verification of Polynomial Identities. J. ACM 27, 4 (Oct. 1980), 701–717. https://doi.org/10.1145/322217.322225
>
> [2] Lingbai Kong, Wengen Li, Hanchen Yang, Yichao Zhang, Jihong Guan, and Shuigeng Zhou. Causalformer: An interpretable transformer for temporal causal discovery. IEEE Transactions on Knowledge and Data Engineering, 37(1):102–115, 2025. doi: 10.1109/TKDE.2024.3484461.